# Can the establishment of National Key Ecological Functional Zones improve air quality?: An empirical study from China

**Guangqin Li** [1]*, **Lingyu Li** [2], **Xing Li** [2], **Yu Chen** [3]

**1** College of International Trade & Economics, Anhui University of Finance & Economics, Bengbu, Anhui, China, **2** Institute of Finance and Economics Research, Shanghai University of Finance and Economics, Shanghai, China, **3** School of East Asian Studies, University of Sheffield, Sheffield, United Kingdom

* zjfcligq@126.com

**Data Availability Statement:** All relevant data are within the manuscript and its Supporting Information files.

**Funding:** We acknowledge the financial support of the Natural Science Foundation of the Zhejiang

## Abstract

Drawing on fine particulate matter ($PM_{2.5}$) from satellite raster data and matching with the county-level socio-economic data from 2008 to 2015 in China, this paper investigates the impacts of the establishment of National Key Ecological Functional Zones (NKEFZ) on environmental quality by employing the difference-in-differences method, which was stablished in June 2011. The results indicate that the establishment of the NKEFZ significantly decreased the concentration values of $PM_{2.5}$, a drop of about 20% during the study period, after the paper controls for other factors affecting air quality. The robustness tests using the maximum and medium concentration values of $PM_{2.5}$ show similar results. Through further analysis, the paper finds that the establishment of NKEFZ can improve the ecological utilization efficiency of land.

## Introduction

Since 1972, Environmental problems have become global problems. As an important global Coordination Conference on environmental governance, United Nations Conference on environment and development (UNCED) is held to discuss global environmental issues and release corresponding policy documents. The countries all over the world have put forward the goal of sustainable development and participated in the process of environmental governance. The developed countries such as the United States have gone through the stage of industrialization and environmental problems have changed from pollution control to environmental behaviour governance. However, China is in the process of development, and its environmental pollution is becoming more and more serious. With the rapid growth of the economy and urbanization, air pollution has become a serious issue in China. Since 2013, China has continuously suffered from serious $PM_{2.5}$ pollution with the average level of fine particulate matter ($PM_{2.5}$) reaching 72 μg/m³; 99.6% of the Chinese population lived in areas with $PM_{2.5}$ exceeding the World Health Organization Air Quality Guideline of 10 μg/m³ [1, 2]. According to the Asian Development Bank Annual Report in 2012, less than 1% of China's 500 largest cities had air quality up to the standards set by the World Health Organization; seven Chinese cities were listed among the ten most polluted cities in the world. Air pollution

Province in China (No. LY18G030014) and the Research Project of Anhui University of Finance and Economics (Green Development Effect of China's Green Credit Policy).

**Competing interests:** The authors have declared that no competing interests exist.

becomes not only a major long-term burden on the Chinese public but also a main obstacle to sustainable development.

The stress on the environment, society and resources is closely related to land use and economic activity [3]. To solve the increasingly urgent problem of environmental protection, China's State Council issued the *National Major Functional Zone Planning* in 2010. Based on population distribution, land use, economic development and urbanization patterns, and the development potential and priorities in different regions, the land is divided into four different functional areas: the optimized development zone, the key development zone, restricted development zone and prohibited development zone. The restricted development zones refer to the zones with weak carrying capacity of resources, poor conditions of large-scale agglomeration economy and population, and is related to the ecological security of the whole country or a large region; the prohibited development zones refer to all kinds of nature protection areas established according to law. Restricted development zones and the prohibited development zones mainly include natural forest protection zones, grassland degradation zones, natural disaster-prone zones, rocky desertification and desertification zones, and zones s with serious soil erosion. According to the requirements of *National Major Functional Zone Planning*, the four function zones should be all adhere to the priority of environmental protection and ecological restoration. Only moderate development, dot-like development, or those characteristic industries suitable for local resources and conditions should be promoted. Environmental protection should guide the orderly transfer of overloaded population and gradually become an important national or regional ecological functional zone. According to the characteristics of restricted development zones, the state named these restricted development zones as national key ecological functional zones (NKEFZ), which are special National Major Functional Zones.

The concept of functional zones in China is similar to those in Western spatial planning [4], where policies influence the spatial expansion of a city to protect environment [5]. Conway & Lathrop [6] used a spatially-explicit model to examine the impacts of future urbanization and alternative land use regulations on the environment before irreversible changes occur. Deboudt et al. [7] found that coastal zone management strategies are useful for natural heritage preservation as land use planning was designed to control urban expansion in coastal areas. Globally, protected areas have been established to preserve natural areas and reduce deforestation [8]. Some of them have made significant contributions to environmental protection [9].

This paper extends the literature on main functional zones and environmental economics in several respects. First, the existing literatures focus on the improvement of environmental quality, mainly from the perspective of environmental policy, few from the perspective of main functional zoning. It is the first paper to apply difference-in-differences model (DID) to examine the impact of NKEFZ policies on air quality in China. Although the policy objective of the NKEFZ policies does not directly affect the improvement of air pollution, it is conducive to the improvement of air quality by changing the production of enterprises and the lifestyle of residents, to adjust the industrial structure in the NKEFZ. Second, the paper uses a new dataset on air pollution derived from satellite raster data on the concentration values of $PM_{2.5}$ from 2008 to 2015. This enables us to evaluate the policy impact on air pollution. This research can expand the related research of environmental economics. Third, previous studies on main functional zones primarily rely on data at the provincial level or prefecture level. The paper uses county level data to construct a panel data, which provides more detailed information at a finer spatial scale. The paper further explores the specific mechanisms through which the designation of the NKEFZ may have influenced air quality in the region.

## Institutional background

The idea of National Major Functional Zones was proposed in the Eleventh Five-Year Plan in China, with the purpose to use the limited land resources efficiently and to improve environmental quality. On the basis of the division of the National Major Functional Zones Plan, China has issued the *National Key Ecological Functional Zones* plan, the planning of *National Key Ecological Functional Zones* (NKEFZ) is introduced in 2010 and officially put into effect in June 2011. A NKEFZ refers to a restricted development zone, which plays a crucial role in protecting the ecological environment by forbidding large-scale high-intensity industrial development. It could include water source protection areas, soil and water erosion control areas, wind break and desertification control areas and biodiversity maintaining areas. Twenty-five areas of Development-Restricted Zones had been identified based on the integrated nationwide assessment, with the total areas of 3.86 million $km^2$, covering 436 county-level administrative regions and accounting for 40.2% of China's territory [10].

The strategy of National Major Functional Zones has gradually become one of the major strategies for China's regional development. This can be seen from the policies issued by the government in recent years. A section is designated to explain what the strategy is at the 12[th] Five-Year Plan for National Economic and Social Development. At the 13[th] Five-Year Plan, another section is designated to explain how to implement the strategy. The concept of NKEFZ was mentioned several times at the 18[th] National Congress of the Communist Party of China (CPC) in 2012. Since the implementation of the "National Main Functional Zones Planning" in 2010, many supporting policies and documents have been issued by the government, covering areas on fiscal arrangement, investment, industrial development, land use, agricultural development, population, ethnicity, environment, and climate change. In the report at the 18[th] Party Congress, the government announced its commitment to accelerating the establishment of eco-compensation mechanism based on the principles including the coordinated development principle, prevention principle, and polluter pays principle.

There have been some studies on functional zones in China. Wu et al. [11] introduced a system dynamic (SD) model to assess land use change in China led by the development priority zoning (DPZ) strategy. By using the Delphi method, a corresponding suitable prioritization of D-U-H-P for the four types of development priority zones, including optimal development zones (ODZ), key development zones (KDZ), restricted development zones (RDZ), and forbidden development zones (FDZ) were identified. Using indicators such as ecological function index, ecological structure index and ecological stress index, Wu et al. [11] analysed ecological changes in the key function zones in China during the years of 2000–2010. Wang et al. [12] presented a system dynamics urban growth model with Yiwu city and Qingtian County as case studies. Their results indicate that the development priority zoning in these two areas influences urban growth in terms of economic growth, migration and land conversion. Li et al. [13] used environmental data from 2009 to 2011 in 37 counties that were granted financial transfer from the government, and found significant correlations between environmental quality and transfer payment funds to the NKEFZ. Fu and Miao [14] argued that the central financial transfer payment to the local ecological function areas should be based on the compensation of the spillover ecological value. Meanwhile, a unified financial transfer payment system should be set up for ecological functional zones across the whole country.

So far existing studies have not evaluated the environmental impacts of the establishment of NKEFZ. This paper employs the following methods and data to identify the environmental effects of the establishment of NKEFZ.

## Methods and data

### Difference-in-differences approach

The difference-in-differences (DID) method is widely used in China's policy evaluation, such as the western development [13]. In the context of the counties which were designated as NKEFZ by the government belong to a "treatment" group and other areas are in a control group. The difference between the treatment group and control group in terms of air quality was a measure of the varying effects of this policy change, after controlling for other factors influencing air quality. The estimation equation follows the following specification:

$$EQ_{it} = \alpha + \beta \cdot T_{it} + \beta \cdot eco_{it} + \delta \cdot T^* eco_{it} + \gamma \cdot X + \mu_i + \nu_t + \zeta_{it} \tag{1}$$

Where $EQ_{it}$ is the air quality of county $i$ at time $t$. $eco$ is a policy dummy variable, which assigned the value of 1 to the counties designated as national key ecological function zones and 0 otherwise. $T$ is the time dummy variable, which equal to 1 from year 2012 to 2015, and 0 from year 2008 to 2011. The $T^* eco$ is the interaction term and $\delta$ was the estimator of the difference-in-differences that examines the effects due to the policy change on local air equality. $X$ is a vector of control variables, and $\gamma$ is a coefficient matrix of control variables. $\mu_i$ is the province fixed effects; $\nu_t$ represents year fixed effects; and $\zeta_{it}$ is the random disturbance term.

The period from 2008 to 2015 is used in the paper because of the following two reasons. First, the policy of National Key Ecological Function Zones was implemented in June 2011. Therefore, the paper uses 2011 as the dividing line to include data before and after it. Second, affected by the economic crisis, $PM_{2.5}$ decreased sharply in 2007, and then returned to the high level and changed smoothly after 2008. Therefore 2008 is chosen as the starting year to avoid other external shocks.

A key assumption of difference-in-differences is known as the "parallel trend" assumption, which supposes that in the absence of treatment, the average outcomes of the treatment group and the control group would follow parallel paths over time. This assumption is tested in the corresponding part of the following text.

### Data and variables

**Dependent variable.** Air pollution is measured by concentration values of $PM_{2.5}$. Data are derived from the satellite information on $PM_{2.5}$ concentration values, provided by the Centre for International Earth Science Information Network (CIESIN) at Columbia University. The dataset contains information on three-year running mean of $PM_{2.5}$ concentration values for grids of 0.01 degree by 0.01 degree (equivalent to 100 $km^2$) since 1998. Adjacent grid points are approximately 10 kilometres apart. For the purpose of the analysis, the paper employed the data from 2008 to 2015 and construct a dataset of $PM_{2.5}$ at the county level. Specifically, for each county-year observation, we calculate the average, median and maximum $PM_{2.5}$ concentration using the data of the grid points that fall within the county [15–18]. It should be pointed out that the satellite data could be affected by meteorological factors in the monitoring process, resulting in its accuracy being slightly lower than the actual monitoring data on the ground. However, $PM_{2.5}$ concentration monitoring sites are sparse and have only been established in recent years. Satellite $PM_{2.5}$ concentration data are useful because they provide long-term monitoring data on $PM_{2.5}$ in China. this paper took the natural logarithm of average $PM_{2.5}$ pollution concentration (*lnavgPM*), median $PM_{2.5}$ pollution concentration (*lnmedPM*) and maximum $PM_{2.5}$ pollution concentration (*lnmaxPM*) to measure air quality. The Table 1 reports the Parallel trends test of three $PM_{2.5}$ pollution concentration. The Difference-in-Differences values of the three variables are significantly negative, which indicates that the establishment of NKEFZ has the effect of improving air quality.

**Table 1. The parallel trends test for the variables.**

| Variables | Before | | | After | | | Difference-in-Differences |
|---|---|---|---|---|---|---|---|
| | Control (C) | Treated (T) | Difference (T-C) | Control (C) | Treated (T) | Difference (T-C) | |
| *lnmean* | 3.568 | 3.174 | -0.394*** | 3.54 | 2.991 | -0.549*** | -0.155*** |
| *lnmed* | 3.563 | 3.17 | -0.392*** | 3.535 | 3.004 | -0.531*** | -0.139*** |
| *lnmax* | 3.743 | 3.393 | -0.350*** | 3.722 | 3.232 | -0.490*** | -0.140*** |
| *lnpgdp* | 9.708 | 9.393 | -0.316*** | 10.192 | 9.947 | -0.245*** | 0.073** |
| *lnpgdp2* | 94.732 | 88.709 | -6.023*** | 104.348 | 99.502 | -4.847*** | 1.233** |
| *agrstr* | 52.294 | 36.909 | -15.385*** | 52.706 | 37.350 | -15.356*** | 0.029 |
| *indstr* | 22.562 | 28.759 | 6.197*** | 19.819 | 25.318 | 5.500*** | -0.697 |
| *lnpopdes* | 45.017 | 36.089 | -8.927*** | 46.172 | 38.082 | -8.090*** | 0.837 |

Notes

"***". "**". "*" Significant at the 1%, 5%, 10% level respectively.

**DID variable.** The difference-in-differences estimator was the independent variable ($eco \times T$). Since the effect of the eco-functional zone policy put into effect in June 2011and has a time lag, 2012 is chosen to reflect the impacts of the policy. The green areas in Fig 1 represent the **NKEFZ**, which are the treatment group in this paper; the red areas are the control group which represent the **non-NKEFZ**. These are 436 county-level administrative regions of the more than 2000 counties in China which belong to the **NKEFZ**.

DID variables include the time dummy and the region dummy: the time dummy variables equal to 1 after the promulgation of the National Key Ecological Function Zones policy; the region dummy variables equal to 1 if the regions belong to the **NKEFZ**. Based on this, the

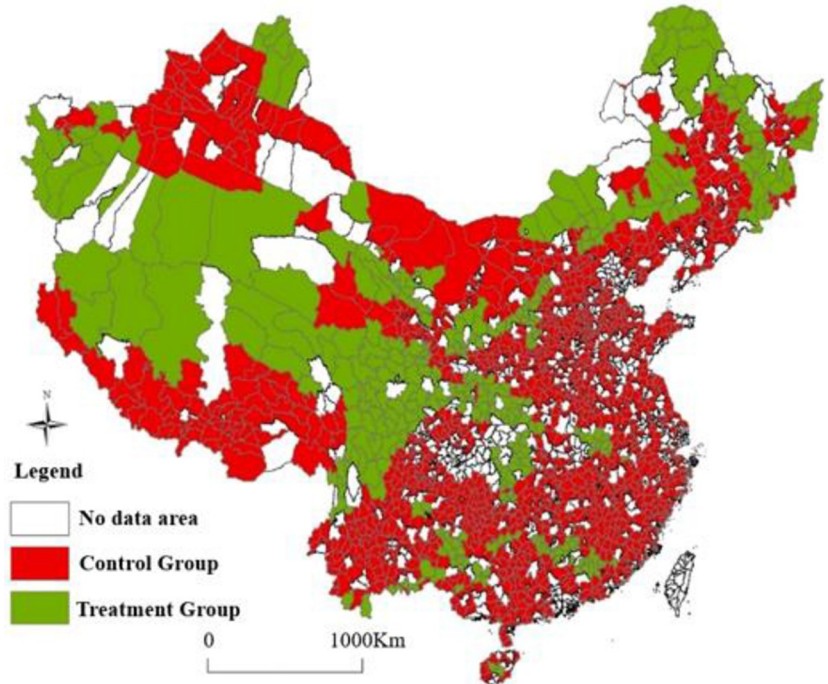

**Fig 1. The map of the NKEFZ and the non-NKEFZ in China.** The map is completed by ArcGIS software. The green areas are the treatment group which the counties belong to the NKEFZ, and the red areas is the control group which the counties belong to the non-NKEFZ. As some counties have no data, they are not included in the study sample, where these areas are shown in white on the map.

interactive items between two dummy variables is DID variable. If the coefficient of the interactive items is negative and significant, it indicates that the air quality of the **NKEFZ** is improved.

**Control variables.** Other control variables were selected to be included in the model based on the "Environmental Impact—Population—Affluence—Technology" equation, which can be expressed as $I = PAT$ and is commonly used to analyse the impacts of human behaviour on the environment. The following variables on population and economic development are included in the model. As technology is relatively stable during the study period, it is excluded from the analysis.

*Regional economic development level* (*lnpgdp*). Log GDP per capita (*lnpgdp*) in counties is used to represent the level of regional economic development. The Environmental Kuznets Curve (EKC) hypothesis describes an inverted U-shaped relationship between environmental quality and economic growth, stating that pollution will first increase with economic growth and then decrease when economic growth reaches a certain turning point [19]. Consistent with the EKC hypothesis, the square of log GDP per capita (*lnpgdp2*) is added to test the non-linear relationship between economic growth and environmental degradation. To exclude the influence of the price factor, the GDP data are adjusted in accordance with the constant 2010 prices. In the Table 1, the Difference-in-Differences value of *lnpgdp* is obviously positive, which indicates that the establishment of NKEFZ is conducive to improving the level of economic development.

*Industrial structure*. The proportions of agricultural and secondary industry outputs in GDP (*agrstr*, *indstr*) are used to measure regional industrial structure. The higher the proportion of secondary industry output in GDP, the more fossil fuels including coal and petroleum might be used, and the more pollution there might be. Thus, a positive coefficient for the proportion of industrial output in GDP and a negative coefficient for the agricultural output in GDP are assumed when modelling air pollution. In the Table 1, the Difference-in-Difference values of *agrstr* and *indstr* aren't significant, which indicates that the establishment of NKEFZ has no effect on improving the industrial structure.

*Population density* (*lnpopdes*). Log Population density, calculated by total population divided by total area, is used instead of population size because there were large variations in coverage areas among counties. According to the IPAT model, air pollution would increase and the quality of the ecological environment would decline, with the increase of population density. A positive relationship between population density and air pollution is hypothesized. In Table 1, the Difference-in-Difference value of (*lnpopdes*) is not significant negative value, which indicates that the establishment of NKEFZ hasn't significant population density reduction effect.

The data on population and economic development from 2008 to 2013 were obtained from the *China Statistical Yearbook for Regional Economy*, and those from 2014 to 2015 are extracted from the *China County Statistical Yearbook*. An overview of the descriptive statistics of these data is provided in Table 1. According to the difference-in-differences test, the three indexes of air quality in the treated group were significantly lower than that of the control group, which shows that the explained variables satisfy the common trend hypothesis. In the control variables, the difference in terms of economic development level between the treatment and control groups is significantly positive only, which mean that the economic development level of the treatment group is significantly lower than that of the treatment group.

## Results

### Benchmark regressions

Table 2 reports the benchmark regression results. Column (1) displays the result after controlling for Year fixed effects and county level fixed effects. Column (2) further includes the

**Table 2. NKEFZ and the environmental quality, baseline estimates.**

| Explanatory variables | Explained Variable: ln*avgPM* | | | |
|---|---|---|---|---|
| | **(1)** | **(2)** | **(3)** | **(4)** |
| *T* | -0.0759*** | -0.0634*** | -0.0650*** | -0.0540*** |
| | (0.0043) | (0.0069) | (0.0046) | (0.0076) |
| *eco* * *T* | -0.1570*** | -0.1531*** | -0.1559*** | -0.1529*** |
| | (0.0052) | (0.0052) | (0.0052) | (0.0052) |
| ln*pgdp* | | -0.2728*** | | -0.2233*** |
| | | (0.0399) | | (0.0429) |
| ln*pgdp2* | | 0.0130*** | | 0.0106*** |
| | | (0.0018) | | (0.0020) |
| *indstr* | | | 0.0017*** | 0.0015*** |
| | | | (0.0003) | (0.0004) |
| *lnpopdes* | | | -0.0002 | 0.0001 |
| | | | (0.0002) | (0.0002) |
| *agrstr* | | | -0.0048*** | -0.0043*** |
| | | | (0.0082) | (0.0091) |
| *_cons* | 3.5068*** | 4.9151*** | 3.7080*** | 4.8355*** |
| | (0.0029) | (0.2151) | (0.0425) | (0.2412) |
| *F* | 269.710 | 221.728 | 203.369 | 174.734 |
| *P Value* | (0.0000) | (0.0000) | (0.0000) | (0.0000) |
| *County fixed* | Y | Y | Y | Y |
| *Year fixed* | Y | Y | Y | Y |
| *Adj. R-squared* | 0.149 | 0.153 | 0.416 | 0.412 |
| *Number of Obs.* | 14071 | 14071 | 14071 | 14071 |

Notes: Due to the multi-collinearity between eco and time fixed effect, the coefficient of eco cannot be estimated, and the estimated coefficient of eco is not reported in the regression table. Standard error of robustness in parentheses

"***". "**". "*" Significant at the 1%, 5%, 10% level respectively.

controls of GDP per capita (*lnpgdp*) and the square of GDP per capita (*lnpgdp2*), to examine whether there is an inverted U-shaped relationship between GDP and pollution. Column (3) adds the controls of industrial structure and population density, while column (4) includes all of the controls. Seeing from Table 2, the difference-in-differences estimators are negative at the 1% significance level, which indicates that the air quality of NKEFZ are significantly improved after the policy was implemented. Among the controls, the coefficients of GDP per capita (column (2) and (4)) are positive while the coefficients of the square of GDP per capita is negative. This is not consistent with the Environmental Kuznets Curve (EKC) hypothesis, which the reason may be that the explained variable is the average $PM_{2.5}$ pollution concentration, not the total pollution level. The coefficients of *indstr* are positive at the 1% significance level, which indicates that a 1% increase in the proportion of industrial output in GDP could increase the average concentrations of $PM_{2.5}$ by 0.2%. The coefficients of *agrstr* are negative at the 1% significance level, which indicates that a 1% increase in the proportion of agriculture output in GDP will reduce the average concentrations of $PM_{2.5}$ by 0.5%. The coefficients of *lnpopdes* are not significant at the 10% significance level, which indicates that population density has no significant effect on average concentrations of $PM_{2.5}$.

## Robustness checks

One concern of the baseline results is that the effects of NKEFZ might be underestimated because of measurement error using average $PM_{2.5}$ concentration values. Alternatively, the paper uses the median and maximum concentrations of $PM_{2.5}$ to measure air pollution. The results are presented in Table 3. Similar to the benchmark regression results, column (1) and (3) reports the results after controlling for the county effects and year effects taking median and maximum $PM_{2.5}$ concentrations as explained variable respectively, while column (2) and (4) further include other controls. The focus is the interactive term $eco^*T$. Its coefficients in columns (1) -(4) are all negative at the 1% significance level, which confirms that air quality in NKEFZ has improved significantly after the policy implementation. The coefficients of $eco^*T$ in column (2) and (4) using median and maximum $PM_{2.5}$ concentrations as explanatory variables are -0.1487 and -0.1651, while the baseline estimate is -0.153 using average $PM_{2.5}$ concentrations (in the column (4) of Table 2). The impact of the policy is significant; the concentrations of $PM_{2.5}$ in NKEFZ decrease about 15% compared to the control group. The coefficients of other control variables are basically consistent with the benchmark regression.

## PSM-DID model

Another concern is that the selection of treatment areas may not be random, that is, counties with better ecological environment and environmental vulnerability might be more likely to

**Table 3. Robust checks.**

| Explanatory variables | Explained Variable: ln*medPM* | | Explained Variable: ln*maxPM* | |
|---|---|---|---|---|
| | (1) | (2) | (3) | (4) |
| T | -0.0723*** | -0.0480*** | -0.0544*** | -0.0333*** |
| | (0.0043) | (0.0076) | (0.0041) | (0.0073) |
| $eco^*T$ | -0.1529*** | -0.1487*** | -0.1691*** | -0.1651*** |
| | (0.0052) | (0.0052) | (0.0050) | (0.0050) |
| lnpgdp | | -0.2109*** | | -0.2227*** |
| | | (0.0430) | | (0.0409) |
| lnpgdp2 | | 0.0100*** | | 0.0106*** |
| | | (0.0020) | | (0.0019) |
| indstr | | 0.0022*** | | 0.0016*** |
| | | (0.0004) | | (0.0003) |
| lnpopdes | | 0.0004* | | 0.0003 |
| | | (0.0002) | | (0.0002) |
| agrstr | | -0.0040*** | | -0.0037*** |
| | | (0.0009) | | (0.0009) |
| _cons | 3.5038*** | 4.7209*** | 3.6861*** | 4.9653*** |
| | (0.0029) | (0.2419) | (0.0028) | (0.2304) |
| F | 253.874 | 166.204 | 284.892 | 184.281 |
| P Value | (0.0000) | (0.0000) | (0.0000) | (0.0000) |
| County fixed | Y | Y | Y | Y |
| Year fixed | Y | Y | Y | Y |
| Adj. R-squared | 0.142 | 0.388 | 0.157 | 0.331 |
| Number of Obs. | 14063 | 14063 | 14054 | 14054 |

Notes: Due to the multi-collinearity between eco and time fixed effect, the coefficient of eco cannot be estimated, and the estimated coefficient of eco is not reported in the regression table. Standard error of robustness in parentheses

"***", "**", "*" Significant at the 1%, 5%, 10% level respectively.

**Table 4. Estimated results of matching.**

| Parter A: Nearest neighbour matching | | | |
|---|---|---|---|
| | *lnavgPM* | *lnmedPM* | *lnmaxPM* |
| *eco* $^*$ *T* | -0.052** | -0.046* | -0.041* |
| | (0.026) | (0.026) | (0.025) |
| Parter B: Radius matching | | | |
| | *lnavgPM* | *lnmedPM* | *lnmaxPM* |
| *eco* $^*$ *T* | -0.497*** | -0.334*** | -0.385*** |
| | (0.016) | (0.016) | (0.016) |
| Parter C: Kernel matching | | | |
| | *lnavgPM* | *lnmedPM* | *lnmaxPM* |
| *eco* $^*$ *T* | -0.236*** | -0.223*** | -0.211*** |
| | (0.014) | (0.010) | (0.017) |
| Parter D: Stratification matching | | | |
| | *lnavgPM* | *lnmedPM* | *lnmaxPM* |
| *eco* $^*$ *T* | -0.062*** | -0.048*** | -0.056*** |
| | (0.020) | (0.012) | (0.018) |

Notes: All the control variables are controlled in all matching methods. Standard error in parentheses
"***". "***". "*" Significant at the 1%, 5%, 10% level respectively.

be designated as NKEFZ. This might result in bias. The paper then employed propensity score match in a difference-in-differences (PSM-DID) model to check the robustness of the results. We compared regions which are designated as NKEFZ (treatment group) with similar areas that are not designated as NKEFZ (control group). Matched control areas are chosen to be similar to the treated NKEFZ in terms of co-variates which might affect both the designation of NKEFZ and the changes of air quality over time. Table 4 displays the results based on the PSM-DID model using the four-matching methods of nearest neighbour matching, radius matching, kernel matching, stratification matching. The results of the four matching methods show that the average treatment effects are significantly negative, which indicates that the establishment of NKEFZ may significantly reduce the average concentration of $PM_{2.5}$ pollutions, the median value of $PM_{2.5}$ pollutions and the maximum concentration of $PM_{2.5}$ pollutions. Because the year and county fixed effects cannot be added to these models, the coefficients of the average treatment effect are some different.

In order to compare the results with the benchmark regression, Table 5 presents the DID regression results after applying the matched samples. In Column (1) and (2), 14022 samples were matched, and the matching rate was 99%. In Column (3) -(6), the matched samples were relatively few, but the matching rate was above 98%. In the six models, the coefficients of *eco* $^*$ *T* were significantly negative, ranging from 0.1486 to 0.1700, which indicates that the establishment of NKEFZ may significantly reduce 14.86%-17% of the average concentration of $PM_{2.5}$ pollutions, the median value of $PM_{2.5}$ pollutions and the maximum concentration of $PM_{2.5}$ pollutions. The estimated results of PSM-DID model still do not support the Environmental Kuznets Curve hypothesis. The proportion of industrial industry and agricultural industry have significant positive and negative effects on $PM_{2.5}$ pollutions respectively.

## Parallel trend test

Policy implementation might have different effects in different periods. Therefore, it is useful to analyse the time effect of policy implementation in the NKEFZ. The specific method is to

**Table 5. Estimated results of PSM-DID model.**

| Explanatory variables | (1) | (2) | (3) | (4) | (5) | (6) |
|---|---|---|---|---|---|---|
| | *lnavgPM* | *lnavgPM* | *lnmedPM* | *lnmedPM* | *lnmaxPM* | *lnmaxPM* |
| *T* | -0.0760*** | -0.0522*** | -0.0728*** | -0.0461*** | -0.0550*** | -0.0321*** |
| | (0.0043) | (0.0077) | (0.0043) | (0.0077) | (0.0041) | (0.0073) |
| $eco^*T$ | -0.1571*** | -0.1528*** | -0.1531*** | -0.1486*** | -0.1700*** | -0.1658*** |
| | (0.0052) | (0.0052) | (0.0052) | (0.0052) | (0.0050) | (0.0050) |
| *lnpgdp* | | -0.2383*** | | -0.2302*** | | -0.2376*** |
| | | (0.0437) | | (0.0438) | | (0.0417) |
| *lnpgdp2* | | 0.0112*** | | 0.0108*** | | 0.0113*** |
| | | (0.0020) | | (0.0020) | | (0.0019) |
| *indstr* | | 0.0015*** | | 0.0023*** | | 0.0017*** |
| | | (0.0004) | | (0.0004) | | (0.0003) |
| *lnpopdes* | | 0.0002 | | 0.0004* | | 0.0003 |
| | | (0.0002) | | (0.0002) | | (0.0002) |
| *agrstr* | | -0.0043*** | | -0.0039*** | | -0.0037*** |
| | | (0.0009) | | (0.0009) | | (0.0009) |
| _cons | 3.5070*** | 4.9202*** | 3.5044*** | 4.8265*** | 3.6865*** | 5.0473*** |
| | (0.0029) | (0.2457) | (0.0029) | (0.2466) | (0.0028) | (0.2348) |
| F | 269.9574 | 175.2305 | 254.4083 | 167.0048 | 287.6338 | 186.6714 |
| *County fixed* | Y | Y | Y | Y | Y | Y |
| *Year fixed* | Y | Y | Y | Y | Y | Y |
| *Adj. R-squared* | 0.1493 | 0.4146 | 0.1421 | 0.3873 | 0.1579 | 0.3350 |
| *Number of Obs.* | 14022 | 14022 | 14054 | 14054 | 14041 | 14041 |

Notes: Due to the multi-collinearity between eco and time fixed effect, the coefficient of eco cannot be estimated, and the estimated coefficient of eco is not reported in the regression table. Standard error of robustness in parentheses

"***". "**". "*" Significant at the 1%, 5%, 10% level respectively.

generate year dummies, named *T2008*, *T2009*, ··, *T2015*; the regional dummy (*eco*) was multiplied by the year dummies to generate eight interactive items names the $eco^*T2008$, $eco^*T2009$, ···, $Eco^*T2015$. Taking $eco^*T2008$ as the reference period, the other seven interactive items were brought into the model (1). The corresponding estimates are reported in Table 6. The air quality in the NKEFZ is better than that in the non NKEFZ, but the impact is significantly negative only after 2012. Comparing the estimated coefficients of $eco^*T2012$, $eco^*T2013$, $eco^*T2014$ and $eco^*T2015$, it is found that these coefficients basically show an upward trend, indicating that with the implementation of the NKEFZ, air quality tends to improve at an accelerating speed. The estimated results in columns (2), (4) and (6) of Table 6 are made into Fig 2. It can be seen that after controlling for other factors, the three explained variables are significantly negative at the level of 5% after 2012, but not significant before 2011, which indicates that the three estimates meet the parallel trend test.

## Mechanism analysis

As we can see from the above results, air quality in NKEFZ improved significantly since the policy was implemented. We further explore the mechanisms for the improvement. Table 7 reports on the impacts of the NKEFZ on economic development and changes in industrial structure. The core explanatory variable is the interactive term $eco^*T$, which represents the treatment effect of the NKEFZ policy on the environment.

**Table 6. Estimated results of time trend analysis.**

| Explanatory variables | (1) | (2) | (3) | (4) | (5) | (6) |
|---|---|---|---|---|---|---|
| | *lnavgPM* | *lnavgPM* | *lnmedPM* | *lnmedPM* | *lnmaxPM* | *lnmaxPM* |
| eco* *T2009* | -0.0103 | -0.0180 | -0.0136 | -0.0112 | -0.0101 | -0.0108 |
| | (0.0104) | (0.0103) | (0.0104) | (0.0104) | (0.0099) | (0.0093) |
| eco* *T2010* | -0.0195 | -0.0153 | -0.0097 | -0.0054 | -0.0161 | -0.0150 |
| | (0.0104) | (0.0103) | (0.0104) | (0.0104) | (0.0099) | (0.0093) |
| eco* *T2011* | -0.0060 | 0.0002 | -0.0094 | -0.0022 | 0.0042 | 0.0107 |
| | (0.0104) | (0.0103) | (0.0104) | (0.0104) | (0.0099) | (0.0096) |
| eco* *T2012* | -0.1472*** | -0.1398*** | -0.1433*** | -0.1351*** | -0.1644*** | -0.1569*** |
| | (0.0104) | (0.0104) | (0.0104) | (0.0104) | (0.0099) | (0.0156) |
| eco* *T2013* | -0.1779*** | -0.1713*** | -0.1776*** | -0.1696*** | -0.1825*** | -0.1756*** |
| | (0.0104) | (0.0104) | (0.0104) | (0.0104) | (0.0099) | (0.0166) |
| eco* *T2014* | -0.1742*** | -0.1671*** | -0.1723*** | -0.1652*** | -0.1818*** | -0.1747*** |
| | (0.0104) | (0.0104) | (0.0104) | (0.0105) | (0.0099) | (0.0154) |
| eco* *T2015* | -0.1969*** | -0.1889*** | -0.1715*** | -0.1632*** | -0.1867*** | -0.1786*** |
| | (0.0104) | (0.0104) | (0.0104) | (0.0105) | (0.0100) | (0.0240) |
| *lnpgdp* | | -0.2294*** | | -0.2315*** | | -0.2366*** |
| | | (0.0439) | | (0.0440) | | (0.0572) |
| *lnpgdp2* | | 0.0109*** | | 0.0109*** | | 0.0112*** |
| | | (0.0020) | | (0.0020) | | (0.0025) |
| *indstr* | | -0.0043*** | | -0.0038*** | | -0.0037*** |
| | | (0.0009) | | (0.0009) | | (0.0014) |
| *lnpopdes* | | 0.0016*** | | 0.0023*** | | 0.0018*** |
| | | (0.0004) | | (0.0004) | | (0.0006) |
| *agrstr* | | 0.0002 | | 0.0004* | | 0.0003 |
| | | (0.0002) | | (0.0002) | | (0.0003) |
| _cons | 3.5074*** | 4.8615*** | 3.5044*** | 4.8218*** | 3.6865*** | 5.0344*** |
| | (0.0029) | (0.2468) | (0.0029) | (0.2477) | (0.0028) | (0.3363) |
| F | 157.9331 | 122.7212 | 147.1846 | 115.6235 | 165.6154 | 85.6495 |
| *County fixed* | Y | Y | Y | Y | Y | Y |
| *Year fixed* | Y | Y | Y | Y | Y | Y |
| *Adj. R-squared* | 0.1525 | 0.3812 | 0.1437 | 0.3592 | 0.1590 | 0.3174 |
| *Number of Obs.* | 14058 | 14058 | 14050 | 14050 | 14041 | 14041 |

Notes: Standard error in parentheses

"***". "**". "*" Significant at the 1%, 5%, 10% level respectively.

The regional economic development, population density and industrial structures tend to improve significantly the average concentrations of $PM_{2.5}$. As showed in Table 7, the coefficients of regional economic development (ln*pgdp*) were significantly positive. But the coefficients of population density (*lnpopdis*) and the proportion of second industrial output in GDP (*indstr*) are significantly negative at the 5% significance level. The coefficients of agriculture output in GDP (*agrstr*) is positive, but not significant. It can be seen from the estimation results that the policies of the NKEFZ policy mainly reduce the proportion of industry and increase the proportion of agriculture, so as to improve the local economic development level, but the NKEFZ policy don't improve the local population density.

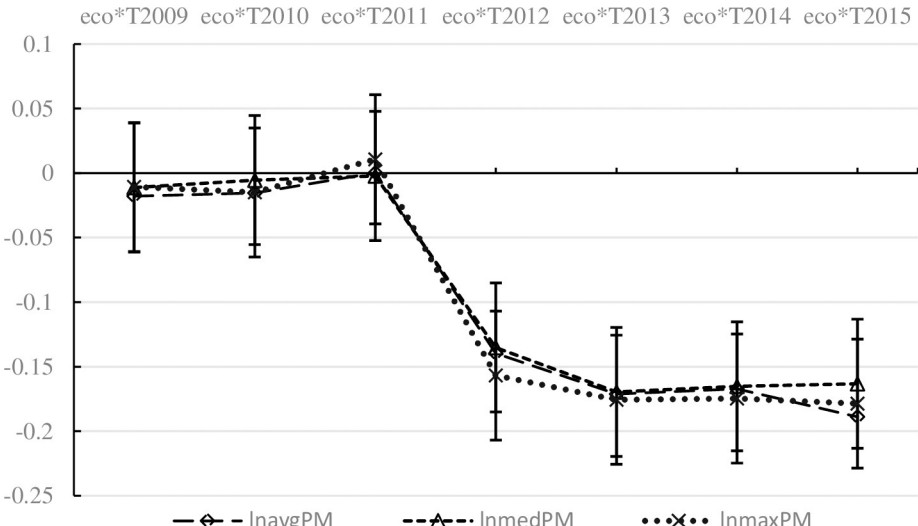

**Fig 2. Parallel trend tests.** Those figures are based on the estimated results in columns (2), (4) and (6) of Table 6, with a confidence level of 5%.

## Conclusion

In this paper we have employed difference-in-differences models and examined the effects of setting up national key ecological functional zones (NKEFZ) on local air quality in China, using satellite data on the concentration values of $PM_{2.5}$ in China's counties from 2008 to 2015. The work is the first to provide empirical evidence on whether and how the policy of establishing national key ecological function zones influences local air quality in the Chinese context. The results show the average concentrations of $PM_{2.5}$ in national key ecological function zones decrease about 20% (approximately 4 μg/m$^3$, the average concentrations of $PM_{2.5}$ is 23 μg/m$^3$) compared with non-national key ecological function zones, after controlling for other factors influencing air pollution. The reduced levels of economic growth, industrial output and population density in NKEFZ provide reasons for the improvement of air quality.

**Table 7. Mechanisms analysis.**

| Explanatory variables | ln*pgdp* | *agrstr* | *lnpopdis* | *indstr* |
|---|---|---|---|---|
| | (1) | (2) | (3) | (4) |
| *eco*$^*$*T* | 0.0718*** | 0.0634 | 0.8282** | -0.6693** |
| | (0.0131) | (0.0818) | (0.4043) | (0.3000) |
| *_cons* | 9.4371*** | 48.9961*** | 42.0165*** | 25.5105*** |
| | (0.0046) | (0.0299) | (0.1497) | (0.1006) |
| *F* | 2.4000 | 44.5865 | 161.0716 | 138.3979 |
| *P Value* | (0.0000) | (0.0000) | (0.0000) | (0.0000) |
| *County fixed* | Y | Y | Y | Y |
| *Year fixed* | Y | Y | Y | Y |
| *Adj. R-squared* | 0.740 | 0.207 | 0.118 | 0.109 |
| *Number of Obs.* | 14059 | 14059 | 14059 | 14059 |

Notes: Standard error of robustness in parentheses

"***". "**". "*" Significant at the 1%, 5%, 10% level respectively.

The significance of this paper is that, on the one hand, it has confirmed that NKEFZ can indeed reduce air pollution, which provides a policy basis for the implementation of China's NKEFZ and other related policies. This paper also provides another governance model for China to further control air pollution and achieve high-quality economic growth. On the other hand, this paper not only provides experience and reference for other developing countries in the treatment of air pollution, but also provides a basis for policy comparison in developed countries such as the United States.

There are a few limitations to the study that are worth emphasizing. On the one hand, this paper only studies the impact of NKEFZ on $PM_{2.5}$, and lacks research on changes in other air pollutants. On the other hand, this paper only studies the effect in the Chinese scenario, and lacks comparative studies with other developed and developing countries. The first problem is limited by the lack of other pollution data in China. At present, the national monitoring data of other pollution in China started in 2013, so it is impossible to identify the impact of NKEFZ in this paper. On the second question, we don't know much about the policies of function zones in other developing countries, and the research on internationalization comparison is relatively insufficient.

## Supporting information

**S1 Data.**
(DTA)

## Author Contributions

**Conceptualization:** Guangqin Li, Yu Chen.

**Data curation:** Guangqin Li, Xing Li.

**Funding acquisition:** Yu Chen.

**Methodology:** Xing Li, Yu Chen.

**Software:** Guangqin Li, Xing Li.

**Supervision:** Yu Chen.

**Writing – original draft:** Lingyu Li.

**Writing – review & editing:** Guangqin Li, Xing Li, Yu Chen.

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
