## [Decision Letter · Decision Letter 0]

17 Jul 2020

PONE-D-20-16975

Can the Establishment of National Key Ecological Functional Zones Improve Air Quality?

——An Empirical Study from China

PLOS ONE

Dear Dr. Li,

Thank you for submitting your manuscript to PLOS ONE. After careful consideration, we feel that it has merit but does not fully meet PLOS ONE’s publication criteria as it currently stands. Therefore, we invite you to submit a revised version of the manuscript that addresses the points raised during the review process.

We look forward to receiving your revised manuscript.

Kind regards,

Bing Xue, Ph.D.

Academic Editor

PLOS ONE

Journal Requirements:

2. Please provide all links to the data bases used in the study in the Methods section.

4. Thank you for stating the following financial disclosure: 'No'

5. We note that Figure 1 in your submission contains map images which may be copyrighted.

We require you to either (a) present written permission from the copyright holder to publish this figure specifically under the CC BY 4.0 license, or (b) remove the figure from your submission:

b. If you are unable to obtain permission from the original copyright holder to publish this figure under the CC BY 4.0 license or if the copyright holder’s requirements are incompatible with the CC BY 4.0 license, please either i) remove the figure or ii) supply a replacement figure that complies with the CC BY 4.0 license. Please check copyright information on all replacement figures and update the figure caption with source information. If applicable, please specify in the figure caption text when a figure is similar but not identical to the original image and is therefore for illustrative purposes only.

6. Please upload a new copy of Figure 1 as the detail is not clear. Please follow the link for more information: https://blogs.plos.org/plos/2019/06/looking-good-tips-for-creating-your-plos-figures-graphics/

7. We note you have included a table to which you do not refer in the text of your manuscript. Please ensure that you refer to Table 7 in your text; if accepted, production will need this reference to link the reader to the Table.

Reviewers' comments:

Reviewer's Responses to Questions

**Comments to the Author**

1. Is the manuscript technically sound, and do the data support the conclusions?

Reviewer #1: Partly

Reviewer #2: Partly

Reviewer #3: Yes

Reviewer #4: Partly

2. Has the statistical analysis been performed appropriately and rigorously? 

Reviewer #1: No

Reviewer #2: Yes

Reviewer #3: N/A

Reviewer #4: No

3. Have the authors made all data underlying the findings in their manuscript fully available?

Reviewer #1: No

Reviewer #2: Yes

Reviewer #3: No

Reviewer #4: No

4. Is the manuscript presented in an intelligible fashion and written in standard English?

Reviewer #1: Yes

Reviewer #2: No

Reviewer #3: No

Reviewer #4: No

5. Review Comments to the Author

Reviewer #1: Comment

Li et al. used satellite image data of air pollutants pm2.5 to investigate whether the policy making would be effective in improving the air environment before and after taking measures. Then, the factors affecting the improvement of the air environment were also investigated. The air environment is generally evaluated using time series analysis, but the authors evaluated the difference between before and after the measures. As a result, when comparing the two points before and after, they conclude that the policy may be useful because the indicator of air pollution has improved considerably.

The improvement of the global air environment is currently entrusted to the United States and China, and it is very meaningful to evaluate the effect of the improvement efforts. This report presents an analysis method that is precise and well stated; however, when evaluated as scientific research, several points of concern arise. Therefore, the authors must improve some aspects of this report before publication.

General

1) Global point of view: Although the report primarily explains China's policies, it lacks an explanation of global trends such as the United Nations and World Health Organization (WHO) policies. Adding this perspective is important because PLOS is an international journal that has a global audience. For example, is the United States now addressing air pollution? Likewise, what are the most recent United Nations Framework Convention on Climate Change guidelines and sustainable development goals? These critical aspects of current air policy should be added to the Introduction or Discussion.

2) Effectiveness: Related to Point (1), the impact of air pollution policies may differ between China and Western countries. It is unlikely that this research will be generalized to the global stakeholders as it is currently written. Furthermore, it is also unlikely that the proposed policy may affect Western countries because a higher degree of freedom is culturally embedded. Although the effects of air pollution policies are important, the discussion is missing the perspective of industry on these policy factors.

3) Sentence structure and Table use: Using several methods for analysis is a good approach for multilateral evaluation. However, if 7 Tables are required for an explanation, then it is necessary to rethink the presentation overall at a structural level. Considering the hypothesis and study in general, the connection from the one Table to the next is weak. The readers would benefit from a streamlined explanation in text so that the arrangement of analysis methods is reasonable and more easily understood.

Major Comments

Introduction

4) The authors describe PM2.5 air pollutants in China in 2013, but the focus of this study is the effect of NKEFZ in 2011. Please explain why PM2.5 air pollutants became a problem after the measures to improve air quality were taken.

5) The pollution level of China is shown in context of the WHO standard, but no information about other countries’ pollution levels was provided. Do the authors have any global information to share?

Methods

6) This study uses the difference-in-differences approach, and the authors should cite and briefly discuss any other studies that have evaluated this issue using the same method.

7) Data are presented using a 3-year moving average, and because the average value is smooth, the fluctuation is moderate. Does this approach underestimate the data when making before/after policy comparisons?

8) The equation called “IPAT” was used in this study, and the authors should explicitly state how data A and data T are quantified.

9) The regional economic development level is evaluated using the Kuznets curve, and the degree of economic development is represented by lnpdgp. Because the approach is comparison of 2 points---before and after---is it possible to also consider the speed of development by region? If so, this aspect would be a valuable addition.

10) Regarding the industrial structure, the secondary industry has a positive impact, but it is speculated that the primary industry may also have an impact of slash and burn, organic dust, and an increase in carbon-containing gas due to livestock production. Is it possible to ignore these effects and call it a negative coefficient?

11) The authors used propensity score matching, but the explanation of the factors used for matching was insufficient. It is desirable to supplement what variables are used for matching and whether the matching was appropriate.

Results

12) In the “Robustness check” part, the authors used the median and maximum data point of PM2.5. Instead, would it be better to use data such as time-weighted average?

Minor Comments

13) In Table1-7, the number of * does not match the significance test result.

14) The Table 7 does not match the description in the text.

Reviewer #2: I read the paper. The paper estimates the effect of National Key Ecological Functional

Zones (NKEFZ) on environmental quality (PM2.5). The issues this paper address is important; hence, the author is given an opportunity of major revision. I attach the comment file.

Reviewer #3: Dear authors. Your work is good and promising. Congratulations.

(1) I thought that you need to revise your manuscript especially by improving English language.

(2) Use reference style that is acceptable by PlosOne journal (i.e., numbering).

(3) Add page numbers and line numbers (this help in reviewing).

(4) Please add a map showing key ecological functions zones in China.

(5) Try to use passive voice when writing (past tense, i.e., reported form).

(6) Avoid or reduce the use of "we" and "our".

(7) How do you explain the low values of Adj.R-squared? (Refer to your Tables - they are very low). Do they have influence to your findings?

(8) See the attachment for few more comments. I have added/deleted some texts. Check line by line.

Reviewer #4: 1. The language and the formatting of the article needs major revisions.

2. The author needs to briefly introduce a terminology when used for the first time in the article, for example - PM 2.5, moderate development, dot-like development etc.

3. In the introduction section, from the second paragraph onward coherence between sentences seem lacking. Even though each sentence adds some new information but the a general flow of thought seems to be lacking in the introduction.

4. The key research questions that the article is trying to address should be mentioned explicitly at the end of introduction section.

5. Description of the content of the article need not be mentioned in the article.

6. In the 'Institutional Background' section, the author might want to give a detailed account of how are NKEFZs different from other regions as it will help the reader to understand the context.

7. Since, the DID model detailed in the manuscript is essentially a regression model, the author should also talk about the assumptions of regression and how they were dealt with.

8. The author should explain why the natural logarithm of PM 2.5 is taken as dependent variable and not the actual PM 2.5 itself.

9. The parallel trend test appears to have been applied incorrectly. The correct method should be testing for difference in PM 2.5 between treatment and control for the years 2008, 2009, 2010, and 2011 individually. Ideally if the data conforms to the parallel trends assumption then the interaction term eco*year (eg. eco*2008) should not be statistically significant.

10. DID models can be easily interpreted using graphs/charts showing the deviance in the treatment line. The author might want to charts in addition to the tables.

11. The EKC hypothesis uses the economic growth as the independent variable and the environmental quality as the dependent variable. However, in the section 3.2.3 the author has concluded that establishment of NKEFZ (independent variable) has lead to improved economic development (dependent variable). Therefore, the conclusion doesn't exactly conform to the EKC hypothesis. Hence, the author should either avoid explaining the model from the EKC hypothesis lens or the author should provide valid arguments with regards to inverse relationship of the EKC curve.

12. At more than one place in the results section, significance level is incorrectly mentioned as 'statistical level'.

13. The author should interpret the results in the manuscript at a pre-defined significance level. At present, results have been interpreted at different significance level (1%, 5%, and 10%).

14. In table 4, which of the variables are dependent and independent are unclear.

15. The paragraph preceding table 5 needs significant revision.

16. In section 4.5, the in the first paragraph table 6 is mentioned instead of table 7.

17. In the conclusion section, the author should discuss about the implication and application of the research. How these results are useful for other geographies. The conclusion section should also talk about the limitations of the study and should provide recommendations to conduct similar studies in future.

Overall Comment:

The idea and analysis framework of the study are good and it appears that the author has worked hard on the study. However, the author needs to make major revisions into the manuscript with regards to the content, language, formatting, and analysis for it to be published in a reputed journal.

Best wishes to the authors.

6. PLOS authors have the option to publish the peer review history of their article (what does this mean?). If published, this will include your full peer review and any attached files.

Reviewer #1: No

Reviewer #2: No

Reviewer #3: No

Reviewer #4: **Yes: **Ankur Kumar

---

## [Author Response · Author response to Decision Letter 0]

28 Oct 2020

Response to Reviewers

First of all, we would like to thank reviewers for your insightful, constructive and helpful comments on our manuscript entitled “Can the Establishment of National Key Ecological Functional Zones Improve Air Quality? —— An Empirical Study from China”. We have carefully considered and addressed all the comments and made necessary revisions in the revised manuscript. We provide a point-by-point response to the reviewers’ comments below.

The points raised by the reviewers are written in bold font, whereas our responses are shown in normal font, and the quotation of the revised manuscript is shown in italic font. 

Reviewer #1:

Li et al. used satellite image data of air pollutants pm2.5 to investigate whether the policy making would be effective in improving the air environment before and after taking measures. Then, the factors affecting the improvement of the air environment were also investigated. The air environment is generally evaluated using time series analysis, but the authors evaluated the difference between before and after the measures. As a result, when comparing the two points before and after, they conclude that the policy may be useful because the indicator of air pollution has improved considerably.

The improvement of the global air environment is currently entrusted to the United States and China, and it is very meaningful to evaluate the effect of the improvement efforts. This report presents an analysis method that is precise and well stated; however, when evaluated as scientific research, several points of concern arise. Therefore, the authors must improve some aspects of this report before publication.

Response:

Thank you so much for your kind, thoughtful and valuable comments. We have done our best to revise the manuscript, but if any additional revision is needed, we will certainly do this in your directions.

Thanks for the referee’s suggestion. In accordance with the requirements of journals and reviewers, we have introduced in detail the international policy comparison of NKEFZ, as well as trends and goals of air pollution-related policies in the Introduction section of the article. In addition, this article also re-emphasizes the application and value of the article at the global level in the Discussion section, making the research meet the interests of the global audience.

General

1) Global point of view: Although the report primarily explains China's policies, it lacks an explanation of global trends such as the United Nations and World Health Organization (WHO) policies. Adding this perspective is important because PLOS is an international journal that has a global audience. For example, is the United States now addressing air pollution? Likewise, what are the most recent United Nations Framework Convention on Climate Change guidelines and sustainable development goals? These critical aspects of current air policy should be added to the Introduction or Discussion.

Response:

Thank you so much for your kind, thoughtful and valuable comments. Following your suggestion, we add some international discussions, In the first chapter of the revised manuscript as follows (please see Page 2-3 of the revised manuscript):

…

Since 1972, Environmental problems have become global problems. As an important global Coordination Conference on environmental governance, United Nations Conference on environment and development (UNCED) is held to discuss global environmental issues and release corresponding policy documents. The countries all over the world have put forward the goal of sustainable development and participated in the process of environmental governance. The developed countries such as the United States have gone through the stage of industrialization and environmental problems have changed from pollution control to environmental behaviour governance. However, China is in the process of development, and its environmental pollution is becoming more and more serious. With the rapid growth of the economy and urbanization, air pollution has become a serious issue in China. Since 2013, China has continuously suffered from serious PM2.5 pollution with the average level of fine particulate matter (PM2.5) reaching 72μg/m3; 99.6% of the Chinese population lived in areas with PM2.5 exceeding the World Health Organization Air Quality Guideline of 10μg/m3 [1,2]. According to the Asian Development Bank Annual Report in 2012, less than 1% of China’s 500 largest cities had air quality up to the standards set by the World Health Organization; seven Chinese cities were listed among the ten most polluted cities in the world. Air pollution becomes not only a major long-term burden on the Chinese public but also a main obstacle to sustainable development.

…

2) Effectiveness: Related to Point (1), the impact of air pollution policies may differ between China and Western countries. It is unlikely that this research will be generalized to the global stakeholders as it is currently written. Furthermore, it is also unlikely that the proposed policy may affect Western countries because a higher degree of freedom is culturally embedded. Although the effects of air pollution policies are important, the discussion is missing the perspective of industry on these policy factors.

Response:

Thank you so much for your kind, thoughtful and valuable comments. As mentioned in the original text, the concept of functional zones in China is similar to the Spatial Planning policy of Western countries (Stull, 1974), where both are through government policies to influence the spatial expansion of a city to protect environment (Wilson et al., 2003). Therefore, NKEFZ is essentially the same as the spatial planning of Western countries. They all plan the type of land use through the government will to achieve the goal of improving air quality. Studying the effect of NKEFZ (or spatial planning) on air pollution in the Chinese scenario can not only show the world on the improvement of air pollution in China as a developing country, but also provide a good reference for other developing countries in the world. More importantly, because China's NKEFZ has the same policy objectives as the spatial planning of western countries, the research conclusions can provide a good comparative study for spatial planning of developed countries. In the meantime, by showing the improvement effect of spatial planning on air pollution under different parameters such as different cultures, different degrees of freedom, and different levels of government intervention, this paper provides a comparison and experience for Western policymakers to better formulate related policies for spatial planning.

Please see Pages 4-5 of the revised manuscript:

…

The concept of functional zones in China is similar to those in Western spatial planning, where policies influence the spatial expansion of a city to protect the environment. Conway & Lathrop used a spatially-explicit model to examine the impacts of future urbanization and alternative land use regulations on the environment before irreversible changes occur. Deboudt et al. found that coastal zone management strategies are useful for natural heritage preservation as land use planning was designed to control urban expansion in coastal areas. Globally, protected areas have been established to preserve natural areas and reduce deforestation. Some of them have made significant contributions to environmental protection.

This paper extends the literature on functional zones and spatial planning in several respects. First, it is the first paper to investigate whether and how the policy of establishing National Key Ecological Functional Zones (NKEFZ) improves air quality in China, using difference-in-differences approaches. It is important to examine the effectiveness of the establishment of ecological functional zones in reducing air pollution because pollution has detrimental impacts on quality of life. So far very few studies have been done on the ecological impacts of functional zones in the Chinese context. We further explore the specific mechanisms through which the designation of NKEFZ may have influenced air quality in the region. Second, we use a new dataset on air pollution derived from satellite raster data on the concentration values of PM2.5 from 2008 to 2015. This enables the paper to evaluate the policy impact on air pollution. Third, previous studies on functional zones primarily rely on data at provincial level. We use county level data to construct a panel data, which provides details at a finer spatial scale.

…

Ref:

Stull, William J. Land Use and Zoning in an Urban Economy. Am Econ Rev. 1974;64(3):337–47. 

Wilson JS, Clay M, Martin E, Stuckey D, Vedder-Risch K. Evaluating environmental influences of zoning in urban ecosystems with remote sensing. Remote Sens Environ. 2003;86(3):303–21. 

3) Sentence structure and Table use: Using several methods for analysis is a good approach for multilateral evaluation. However, if 7 Tables are required for an explanation, then it is necessary to rethink the presentation overall at a structural level. Considering the hypothesis and study in general, the connection from the one Table to the next is weak. The readers would benefit from a streamlined explanation in text so that the arrangement of analysis methods is reasonable and more easily understood.

Response:

Thank you very much for your kind, thoughtful and valuable comments. This paper mainly attempts to use these 7 tables to conduct a more detailed study on the impact of NKEFZ on air quality, so that the results of the study are innovative and robust. According to the reviewer’s suggestions, we have reorganized the logical structure, research process, and language expression of the full text based on hypothesis and study, making the research process more rigorous, logical structure more reasonable.

4) The authors describe PM2.5 air pollutants in China in 2013, but the focus of this study is the effect of NKEFZ in 2011. Please explain why PM2.5 air pollutants became a problem after the measures to improve air quality were taken.

Response:

Thank you very much for your kind, thoughtful and valuable comments. The PM2.5 pollution in 2013 is the most serious pollution phenomenon in the history of China, such as the smog event experienced in Britain for 60 years. The main purpose of this paper is to introduce the research problem of PM2.5 pollution control. The NKEFZ policy implemented by the Chinese government in 2011 aims to divide the national land into difference functional regions and implement difference industrial development policies. Therefore, this policy can be used as a quasi-natural experiment of pollution control.

In our paper, the analysis of this problem is not very clear, which leads to the reviewers don’t understand the two issues. In my revised manuscript, we have added a description of China's PM2.5 pollution situation before 2011 in the first paragraph, and through the comparison of PM2.5 concentrations before and after the implementation of the NKEFZ policy, we have more vividly demonstrated the inhibitory effect of NKEFZ on air pollution. Please see Page 5 of the revised manuscript:

…

The idea of "National Major Functional Zones" was proposed in the Eleventh Five-Year Plan in China, with the purposes to use the limited land resources efficiently and to improve environmental quality. The planning of "National Key Ecological Functional Zones" is introduced in 2010 and officially put into effect in June 2011. A NKEFZ refers to a restricted development zone, which plays a crucial role in protecting the ecological environment by forbidding large-scale high-intensity industrial development. It could include water source protection areas, soil and water erosion control areas, wind break and desertification control areas and biodiversity maintaining areas. Twenty-five areas of Development-Restricted Zones had been identified based on integrated nationwide assessment, with the total areas of 3.86 million km2, covering 436 county-level administrative regions and accounting for 40.2% of China’s territory.

…

5) The pollution level of China is shown in context of the WHO standard, but no information about other countries’ pollution levels was provided. Do the authors have any global information to share?

Response:

Thank you very much for your kind, thoughtful and valuable comments. We have collected pollution information about other countries from World Health Organization, World Bank databases, satellite data. In our first paragraph, we have a comparative analysis of China's urban pollution ranking in global urban pollution. The PM2.5 pollution data used in this study is provided by the Centre for International Earth Science Information Network (CIESIN) at Columbia University, which provides PM2.5 pollution data for each country in the world. Please refer to the website: https://sedac.ciesin.columbia.edu/#

6) This study uses the difference-in-differences approach, and the authors should cite and briefly discuss any other studies that have evaluated this issue using the same method.

Response:

Thank you very much for your kind, thoughtful and valuable comments. Based on your suggestions, we add a part of the explanation in part of Difference-in-differences approach. Please see page 7 of the revised manuscript.

…

As a policy evaluation method, the difference-in-differences (DID) method is commonly used in social science research. Specifically, the implementation of a policy is only affected in some regions (some countries or some regions within the country), while other regions are not affected by the policy. By comparing the differences of policy implementation areas before and after, and comparing the the differences with other regions that are not affected by policy shocks, the net effects of policy shocks on policy implementation regions can be identified. This method takes a policy shock in social science as a quasi-natural experiment and is widely used as a policy evaluation method.

…

7) Data are presented using a 3-year moving average, and because the average value is smooth, the fluctuation is moderate. Does this approach underestimate the data when making before/after policy comparisons?

Response:

Thank you very much for your kind, thoughtful and valuable comments. In the part of “trend analysis of policy impact”, we do not use the three-year moving average method to study. It is assumed that the policy shock occurred in the previous year, so that the dummy variable of the previous year is set to 1, otherwise it is 0. Similar settings are also adopted for dummy variables of other years. These dummy variables are independent of each other. If these dummy variables are introduced into the regression model, the estimated results will not affect each other. Therefore, we believe that there is no bias in the estimation results.

8) The equation called “IPAT” was used in this study, and the authors should explicitly state how data A and data T are quantified.

Response:

Thank you very much for your kind, thoughtful and valuable comments. Based on your suggestions, we add a part of the explanation in part of Difference-in-differences approach. Please see page 11 of the revised manuscript.

…

The following variables lnpopdes represents on population (P), the lnpgdp and lnpgdp2 represent economic development (A), agrstr and indstr represent technical level (T), it is excluded from the analysis.

…

9) The regional economic development level is evaluated using the Kuznets curve, and the degree of economic development is represented by lnpdgp. Because the approach is comparison of 2 points---before and after---is it possible to also consider the speed of development by region? If so, this aspect would be a valuable addition.

Response:

Thank you very much for your kind, thoughtful and valuable comments. Your suggestion is very good. In the sample we studied, the areas of the treatment group are the areas with national ecological function areas, which the economic development level of these areas is not high. If following your ideas, the results will lead to very few samples of treatment groups entering high development areas, which will lead to some errors in the estimation results. 

Because the problem we are considering is the treatment effect of NKEFZ policy, which is an average effect. The treatment effect is of great importance to the implementation of the NKEFZ policy, but the significance of sub sample may not be very important.

10) Regarding the industrial structure, the secondary industry has a positive impact, but it is speculated that the primary industry may also have an impact of slash and burn, organic dust, and an increase in carbon-containing gas due to livestock production. Is it possible to ignore these effects and call it a negative coefficient?

Response:

Thank you very much for your kind, thoughtful and valuable comments. 

First of all, as the reviewer said, slash and burn, organic dust, and an increase in carbon-containing gas due to livestock production in the primary industrial process can indeed have an impact on air pollution in theory. However, these impacts are not large or even negligible compared to the impact of secondary industry on air pollution in reality. Because the primary industry has little influence, the relevant literature is also less involved in the selection and research of these variables.

Secondly, from the perspective of econometrics, this article has made every effort to collect data and added major variables that can theoretically have a significant impact on PM2.5. In the meantime, the random disturbance term of formula (1) contains the impact of all unobserved variables on PM2.5, including slash and burn, organic dust, and an increase in carbon-containing gas due to livestock production, etc., which can directly affect PM2.5 but have little effect, and all other direct or indirect effects.

Finally, as a variable in the primary industry that can have a major impact on PM2.5, the negative coefficient of the proportions of agricultural industry outputs in GDP has been theoretically explained. On the other hand, it has also been proved in the quantitative regression below, so in general, it is possible to ignore these effects and call it a negative coefficient.

11) The authors used propensity score matching, but the explanation of the factors used for matching was insufficient. It is desirable to supplement what variables are used for matching and whether the matching was appropriate.

Response:

Thank you very much for your kind, thoughtful and valuable comments. In the process of PSM-DID, we control all the control variables and directly use the diff command of Stata. This command only reports the results and does not report the estimated results of other control variables. The number of matched samples is reported in the paper. See the page 18 of the revised manuscript.

…

When matching, all control variables are controlled.

In order to compare the results with the benchmark regression, Table 5 presents the DID regression results after applying the matched samples. in Column (1) and (2), 14022 samples were matched, and the matching rate was 99%. in Column (3) -(6), The matched samples were relatively few, but the matching rate was above 98%.

…

12) In the “Robustness check” part, the authors used the median and maximum data point of PM2.5. Instead, would it be better to use data such as time-weighted average?

Response:

Thank you very much for your kind, thoughtful and valuable comments. The median value of PM2.5 pollution is most similar to the average value of PM2.5 pollution, so we can see whether the results are robust. The maximum value is utilized to explain whether the policy impact will reduce the pollution severity of the treatment group area. The PM2.5 value with time weighting may be better, but the data we use is the current year value provided by CIESIN, so time weighting is not available.

13) In Table1-7, the number of * does not match the significance test result.

Response:

Thank you very much for your kind, thoughtful and valuable comments. Thanks for your corrections. We have reviewed the full text and corrected the relevant significance level.

14) The Table 7 does not match the description in the text.

Response:

Thank you very much for your kind, thoughtful and valuable comments. The original number was wrong, which caused misunderstanding by the reviewers. We have revised the numbering of Table 7 in the Mechanism analysis section, and further corrected the interpretation of Table 7. Please see pages 21-22 of the revised manuscript.

…

We have known that regional economic development, population density and industrial structures tend to improve significantly the average concentrations of PM2.5. As shown in Table 7, the coefficients of regional economic development (lnpgdp) were significantly positive. But the coefficients of population density (lnpopdis) and the proportion of second industry output in GDP (indstr) are significantly negative at the 5% significance level. the coefficients of agriculture output in GDP (agrstr) is positive, but not significant. It can be seen from the estimation results that the policies of the NKEFZ policy mainly reduce the proportion of industry and increase the proportion of agriculture, so as to improve the local economic development level, but the NKEFZ policy don’t improve the local population density.

…

Once again, we would like to thank you for your thoughtful and valuable comments and suggestions. If you have any further requests, please do not hesitate to contact us so that we can address your further concerns.

 

Reviewer #2: 

I read the paper. The paper estimates the effect of National Key Ecological Functional Zones (NKEFZ) on environmental quality (PM2.5). The issues this paper address is important; hence, the author is given an opportunity of major revision. I attach the comment file.

Response:

Thank you very much for your kind, thoughtful and valuable comments. We have done our best to revise the manuscript, but if any additional revision is needed, we will certainly do this in your directions.

1. The existing title doesn't provide a complete picture of the study and need revision.

Response:

Thank you very much for your kind, thoughtful and valuable comments. The title is “Can the Establishment of National Key Ecological Functional Zones Improve Air Quality? —— An Empirical Study from China”. I want to study the relationship between the establishment of National Ecological Functional Zones and air quality in China. The questions used in the title may express partial information. We can revise it is: Establishment of National Key Ecological Functional Zones and air quality —— An Empirical Study from China. However, we are still not sure whether this title is OK or not, but if any additional revision is needed, we will certainly do so under your directions.

2. Abstract:

2.1. Include the current PM2.5 level with reference to the standard level so that it will be easier for the reader to understand the alarming situation of air pollution and also the relevance of the study. 

2.2. Mention when NKEFZ was established in the abstract itself.

2.3. The last sentence of the abstract needs revision to reflect the mechanism through which NKEFZ reduced the PM2.5. The sentence is written in a way to present the result of the effect of NKEFZ on land use.

Response:

Thank you very much for your kind, thoughtful and valuable comments. We have done our best to revise the abstract. Please see page 2 of the revised manuscript.

…

Drawing on fine particulate matter (PM2.5) from satellite raster data and matching with the county-level socio-economic data from 2008 to 2015 in China, this paper investigates the impacts of the establishment of National Key Ecological Functional Zones (NKEFZ) on environmental quality by employing the difference-in-differences method, which was stablished in June 2011. The results indicate that the establishment of the NKEFZ significantly decreased the concentration values of PM2.5, a drop of about 20% during the study period, after we control for other factors affecting air quality. The robustness tests using the maximum and medium concentration values of PM2.5 show similar results. Through further analysis, we find that establishment of NKEFZ can improve the ecological utilization efficiency of land.

…

Introduction:

2.4. There is a discussion of the restricted zones, which is one of the four zones classified by the National Major Functional Zone Planning. Describing the criteria used to classify helps the reader about the policy source for NKEFZ. 

Response:

Thank you very much for your kind, thoughtful and valuable comments. The criteria mentioned in the original text is based on population distribution, land use, economic development and urbanization patterns, and the development potential and priorities in different regions. By further reviewing the relevant policy documents of the main functional zones, we found that the main functional zones are divided according to the resource and environment carrying capacity of different regions. Different regions are divided into different regional spatial units with specific main functions based on their spatial differentiation of natural environmental factors, social and economic development levels, ecosystem characteristics, and human activity forms. For example: For regions with strong comprehensive strength, large economic scale, capable of supporting and driving the national economic development, strong scientific and technological innovation strength, capable of leading and driving national independent innovation and structural upgrading, the state defines it as an optimized development zone. For areas with strong resource and environmental carrying capacity and a certain development space, they will be regarded as key development zones, and industrialization and urbanization will be mainly implemented. For areas that have strong comprehensive agricultural production capacity, or undertake important ecological functions such as water conservation, soil and water conservation, wind prevention and sand fixation, and biodiversity maintenance, the state defines it as a restricted development zone. Among them, the restricted development zone related to national ecological security is the NKEFZ studied in this paper. For areas such as nature reserves, scenic spots, forest parks, and geological parks, the state defines them as restricted development areas.

Please see pages 3-4 of the revised manuscript.

…

The restricted development zones refer to the zones with weak carrying capacity of resources, poor conditions of large-scale agglomeration economy and population, and is related to the ecological security of the whole country or a large region; the prohibited development zones refer to all kinds of nature protection areas established according to law. Restricted development zones and the prohibited development zones mainly include natural forest protection zones, grassland degradation zones, natural disaster-prone zones, rocky desertification and desertification zones, and zones s with serious soil erosion.

…

2.5. At the end of the first page, the author writes "They should adhere to the priority of environmental protection and ecological restoration. Only moderate development, dot-like development, or those characteristic industries suitable for local resources and conditions should be promoted. Environmental protection should guide the orderly transfer of overloaded population and gradually become an important national or regional ecological functional zone.", Is this the author's view or description of the policy documents? The description needs revision with appropriate reference.

Response:

Thank you very much for your kind, thoughtful and valuable comments. See the page 3-4 of the revised manuscript.

…

According to the requirements of National Major Functional Zone Planning, the four function zones should be all adhere to the priority of environmental protection and ecological restoration.

…

2.6. It seems NKEFZ is restricted development zone—named differently, yes? If so, please provide reference and reason why it is a specialized functional zone. 

Response:

Thank you very much for your kind, thoughtful and valuable comments. Your question is very good. On the basis of the division of the National Major Functional Zones Plan, China has issued the National Key Ecological Functional Zones plan, taking some zones as the key areas proposed promoting the construction of ecological civilization system. As of 2016, 676 counties have entered the National Key Ecological Functional Zones, accounting for 53% of the total land area. Please see page 5 of the revised manuscript.

…

On the basis of the division of the National Major Functional Zones Plan, China has issued the National Key Ecological Functional Zones plan, the planning of National Key Ecological Functional Zones (NKEFZ) is introduced in 2010 and officially put into effect in June 2011.

…

2.7. First, it is the first paper to investigate whether and how the policy of establishing National Key Ecological Functional Zones (NKEFZ) improves air quality in China, using difference-in-difference approaches. Is this paper the first one to a) investigate the NKEFZ effect or b) use DID to investigate the NKEFZ?

Response:

Thank you very much for your kind, thoughtful and valuable comments. Ibe so sorry that the expression in the manuscript has caused you to make an ideal deviation. We have revised the manuscript, please see the page 5 of the revised manuscript.

…

First, it is the first paper to apply difference-in-differences mothed (DID) to examine the impact of NKEFZ policies on air quality in China.

…

2.8. If used for one or very few time, please use full forms like D-U-H-P, DPZ, SD and others.

Response:

Thank you very much for your advice. Many abbreviations are used in this manuscript because these expressions can be used many times.

2.9. NKEFZ is mentioned here and there. A separate paragraph that describes NKEFZ, its classification, programs, and other detail is helpful to understand the (NKEFZ) program being investigated.

Response:

Thanks for your suggestion. We have revised the full text and use a separate paragraph to describe NKEFZ, its classification, programs, and other details in Institutional background. Please see pages 5-7 of the revised manuscript.

…

Institutional background

The idea of National Major Functional Zones was proposed in the Eleventh Five-Year Plan in China, with the purpose to use the limited land resources efficiently and to improve environmental quality. On the basis of the division of the National Major Functional Zones Plan, China has issued the National Key Ecological Functional Zones plan, the planning of National Key Ecological Functional Zones (NKEFZ) is introduced in 2010 and officially put into effect in June 2011. A NKEFZ refers to a restricted development zone, which plays a crucial role in protecting the ecological environment by forbidding large-scale high-intensity industrial development. It could include water source protection areas, soil and water erosion control areas, wind break and desertification control areas and biodiversity maintaining areas. Twenty-five areas of Development-Restricted Zones had been identified based on the integrated nationwide assessment, with the total areas of 3.86 million km2, covering 436 county-level administrative regions and accounting for 40.2% of China’s territory [10].

The strategy of National Major Functional Zones has gradually become one of the major strategies for China's regional development. This can be seen from the policies issued by the government in recent years. A section is designated to explain what the strategy is at the 12th Five-Year Plan for National Economic and Social Development. At the 13th Five-Year Plan, another section is designated to explain how to implement the strategy. The concept of NKEFZ was mentioned several times at the 18th National Congress of the Communist Party of China (CPC) in 2012. Since the implementation of the "National Main Functional Zones Planning" in 2010, many supporting policies and documents have been issued by the government, covering areas on fiscal arrangement, investment, industrial development, land use, agricultural development, population, ethnicity, environment, and climate change. In the report at 18th Party Congress, the government announced its commitment to accelerate the establishment of eco-compensation mechanism based on the principles including the coordinated development principle, prevention principle, and polluter pays principle. A restricted industry access system in the NKEFZ is required at the Fifth Plenary Meeting of the CPC. 

There have been some studies on functional zones in China. Wu et al.[11] introduced a system dynamic (SD) model to assess land use change in China led by the development priority zoning (DPZ) strategy. By using the Delphi method, a corresponding suitable prioritization of D-U-H-P for the four types of development priority zones, including optimal development zones (ODZ), key development zones (KDZ), restricted development zones (RDZ), and forbidden development zones (FDZ) were identified. Using indicators such as ecological function index, ecological structure index and ecological stress index, Wu et al. [11] analyses ecological changes in the key function zones in China during the years of 2000-2010. Wang et al. [12] presents a system dynamics urban growth model with Yiwu city and Qingtian County as case studies. Their results indicate that the development priority zoning in these two areas influences urban growth in terms of economic growth, migration and land conversion. Li et al. [13] used environmental data from 2009 to 2011 in 37 counties that were granted financial transfer from the government, and found significant correlations between environmental quality and transfer payment funds to the NKEFZ. Fu and Miao [14] argued that the central financial transfer payment to the local ecological function areas should be based on the compensation of the spillover ecological value. Meanwhile, a unified financial transfer payment system should be set up for ecological functional zones across the whole country. 

So far existing studies have not evaluated the environmental impacts of the establishment of NKEFZ. This paper employs the following methods and data to fill in the gap.

…

3. Method and Data:

3.1. Add paragraph describing the intervention and its implementation. For example, after the establishment of NKEFZ, what happened to industry and other development activities—that has a relationship with air pollution? Where did they shift to other places (which is the control area in the study)?

Response:

Thank you very much for your kind, thoughtful and valuable comments. Based on the data, we can see the change of industrial structure from the descriptive statistical table in Table 1. However, we can't see where the industry is going. Therefore, we did not compare the data before and after the establishment of NKEFZ in the manuscript.

3.2. In section 3.1, in the third paragraph provide few previous strategies that have used similar approach or estimation strategy (using longitudinal data at country or area as a DID to estimate the effect of policy intervention). 

Response:

Thank you very much for your kind, thoughtful and valuable comments. According to your suggestion. We have provided related similar revise. Please see the page 7 of the revised manuscript.

…

The difference-in-differences method is widely used in China's policy evaluation, such as the western development [1].

…

Ref:

[1] Jia, J., Ma, G., Qin, C., & Wang, L., 2020. Place-based policies, state-led industrialisation, and regional development: Evidence from China's Great Western Development Programme. Europ. Econ. Rev., (123): 1-21.

3.3. Mention a few recent studies that have used PM2.5 data from the Centre for International Earth Science Information Network (CIESIN) at Columbia University. 

Response:

Thanks for the referee’s kind advice. We have provided related a similar revision. Please see page 7 of the revised manuscript. 

…

Specifically, for each country-year observation, we calculate the average, median and maximum PM2.5 concentration using the data of the grid points that fall within the county [1-3].

…

Ref:

[2] Boys BL, Martin R V., Van Donkelaar A, MacDonell RJ, Hsu NC, Cooper MJ, et al. Fifteen-year global time series of satellite-derived fine particulate matter. Environ Sci Technol. 2014;48(19):11109–18. 

[3] Van Donkelaar A, Martin R V., Spurr RJD, Burnett RT. High-Resolution Satellite-Derived PM2.5 from Optimal Estimation and Geographically Weighted Regression over North America. Environ Sci Technol. 2015;49(17):10482–91. 

[4] van Donkelaar A, Martin R V., Brauer M, Hsu NC, Kahn RA, Levy RC, et al. Global Estimates of Fine Particulate Matter using a Combined Geophysical-Statistical Method with Information from Satellites, Models, and Monitors. Environ Sci Technol [Internet]. 2016 Apr 5;50(7):3762–72. Available from: https://pubs.acs.org/doi/10.1021/acs.est.5b05833

3.4. How big is the county as compared to grid value? Many counties in one grid or many grids on one county? How was the average/median calculated at the county level? When grid falls in two counties how was it taken care or resolved? 

Response:

Thank you very much for your kind, thoughtful and valuable comments. Your question is very good. The size of a grid (0.01*0.01) is equivalent to but not equal to a square kilometer, so the area of a county is much larger than the size of a grid. Because the boundary of the county is not regular geometry, it is likely that a grid is on the boundary of two counties. In this case, the data of this grid will enter the two counties respectively. Then the average value is computed according to the amount of raster data each county has. The calculation process is completed by ArcGIS software.

3.5. Figure 1; Is the map used from somewhere else? If yes, reference the source. What is the figure on the bottom-right side? Use a clear figure.

Response:

Thank you very much for your kind, thoughtful and valuable comments. Figure 1 is based on our own data through ArcMap. The text on the bottom right is not very clear. We add a note on the edge. Please see page 7 of the revised manuscript. 

…

Fig 1. Study region

Note: According to the author's own data, the author made it through ArcMap.

…

3.6. Figure on PM2.5 by country (if possible) would help the reader to understand air pollution condition in a different location. 

Response:

Thank you very much for your kind, thoughtful and valuable comments. Your question is very good. We try to show the PM2.5 of each county in the diagram, but when it is implemented by ArcMap. It is not possible to display the two indicators at the same time (PM2.5 value and whether it is a treatment group). Therefore, we have not dealt with this issue well.

4. Result:

4.1. Write the unit and measurement of the dependent variable. 

Response:

Thank you very much for your kind, thoughtful and valuable comments. The calculation method and specific economic meaning of each variable have been introduced in the variable description section. Therefore, the unit and measurement of variables are not marked in this part of the result.

4.2. A figure showing the mean difference between PM2.5 (before-after and with and without the intervention) enables the reader to understand the effect of intervention easily.

Response:

Thank you very much for your kind, thoughtful and valuable comments. In the part of time trend analysis and dynamic analysis, it is actually a parallel trend test. Its function is to analyze the differences before and after the implementation of the policy. Therefore, we did not draw a graph to illustrate the parallel trend test.

4.3. Clearly describe the matching criteria (for result presented in Table 4).

Response:

Thank you very much for your kind, thoughtful and valuable comments. Table 4 shows the DID estimation after matching by four matching methods. In view of these matching methods, we think its theoretical basis is relatively introduced in the classic textbooks, we mainly use it, so there is no method description in the manuscript.

4.4. Is this "As shown in table 6, the coefficients." in the paragraph just before Table 7 refereeing to Table 6 or Table 7? A sentence in the (same) paragraph just before Table 7 reads "But the coefficients of population density (lnpopdis) and the proportion of second industry output in GDP (indstr) are significantly negative at 5% significance levels." But the estimates are different when compared to Table-6 or Table-7.

Response:

Thank you very much for your kind, thoughtful and valuable comments. Your rigorous attitude towards research is worth learning. In the process of writing and revising, there are many small problems due to the strict examination of the text of the paper. All the analyses in the Mechanism analysis are the description and explanation of Table 7. We have made changes in the relevant places in the revised manuscript.

4.5. Add short discussion that relates to the effect of the similar or relevant intervention on air pollution in China or other parts of the china or world, if available. 

Response:

Thank you very much for your kind, thoughtful and valuable comments. Your rigorous attitude towards research is worth learning. In the process of writing and revising, there are many small problems due to the strict examination of the text of the paper. All the analyses in the Mechanism analysis are the description and explanation of Table 7. We have made changes in the relevant places in the revised manuscript.

5. Conclusion:

5.1. The use of NKEFZ and its full form is confusing. Use NKEFZ with full form first time and continue to use NKEFZ. 

Response:

Thank you very much for your kind, thoughtful and valuable comments. We have revised all the abbreviations and full names as requiring by the reviewers. We use full form and abbreviations when we first use terminology, and abbreviations in subsequent content. We also listed a comparison table of abbreviations and full names

5.2. Include mean PM2.5 in non NKEFZ area so that reader can understand what 20% reduction refers indicate. 

Response:

Thank you very much for your kind, thoughtful and valuable comments. We have the corresponding part of the conclusion. Please see page 23 of the revised manuscript. 

…

Our results show the average concentrations of PM2.5 in national key ecological function zones decrease about 20% (approximately 4 ug/m3, the average concentrations of PM2.5 is 23 ug/m3) compared with non-national key ecological function zones, after controlling for other factors influencing air pollution.

…

B. Overall comment:

1. The paper needs revision through-out the manuscript for clarity and flow. An example is breaking the long sentence in the second sentence of the Introduction into two sentence 

2. Recheck to improve typos and another formatting, for example in section 3.2.2 "The green areas in" has a different font.; paragraph break in 4.1 

3. Use standard note for significance level in all tables.

Response:

Thank you very much for your patience. We have revised the full text and modified the existing grammar and other low-level errors. Of course, if any additional revision is needed, we will certainly do so under your directions.

 

Reviewer #3: 

Dear authors. Your work is good and promising. Congratulations.

(1) I thought that you need to revise your manuscript especially by improving English language.

Response:

Thank you very much for your kind, thoughtful and valuable comments. We have done our best to revise the manuscript, but if any additional revision is needed, we will certainly do so under your directions.

(2) Use reference style that is acceptable by Plos One journal (i.e., numbering).

Response:

Thank you very much for your kind, thoughtful and valuable comments. Depending on the requirements of the Plos One journal, we have made a comprehensive revision of the manuscript, including references.

(3) Add page numbers and line numbers (this help in reviewing).

Response:

Thanks again for your kind advice, we have added page numbers and line numbers in the revised manuscript.

(4) Please add a map showing key ecological functions zones in China.

Response:

Thanks for the referee’s suggestion. The red part in the Fig 1 is the National Key Ecological Functions Zones in China. Since the sample in this paper is county-level, the county-level is also accurate in the map, and the administrative boundaries between counties are drawn.

(5) Try to use passive voice when writing (past tense, i.e., reported form).

Response:

Thank you very much for your kind, thoughtful and valuable comments. According to your suggestion, we have rewritten in accordance with your requirements.

(6) Avoid or reduce the use of "we" and "our".

Response:

Thank you very much for your kind, thoughtful and valuable comments. We have reorganized the language and corrected the expression

(7) How do you explain the low values of Adj. R-squared? (Refer to your Tables - they are very low). Do they have influence to your findings?

Response:

Thank you very much for your kind, thoughtful and valuable comments. There are numerous aspects to the explanation of Adj. R-squared has low values:

First of all, the sample size of this article is huge. Compared with other small sample studies at the provincial level, this article studied samples of 436 county-level administrative regions in China from 2008 to 2015, with a sample size of 3488 (436*8). The huge sample size reduces the Adj. R-squared in the regression, which is reasonable (for example, when there are only two variables and two samples, the Adj. R-squared obtained by the regression is equal to 1, because two points determine a straight line, and as the sample size gradually increases, Adj. R-squared will gradually become smaller).

Secondly, China has a very large area, and there are huge differences in various factors between different regions. These samples are scattered in various places in China, and the variance between various variables is also very large. Therefore, Adj. R-squared has low values are also reasonable.

Finally, low values of Adj. R-squared will not affect our findings. In the measurement model, regression is divided into explanatory regression and predictive regression. Predictive regression generally pays more attention to R-squared explanatory regression pays more attention to the overall significance of the model, the statistical significance of the independent variables and the significance of economic significance. This paper belongs to an explanatory regression, so the conclusions obtained are still significant and robust. 

(8) See the attachment for few more comments. I have added/deleted some texts. Check line by line.

Response:

Thank you very much for your kind, thoughtful and valuable comments. We have read your attachment in detail and revised it one by one.

Once again, we would like to thank you for your thoughtful and valuable comments and suggestion. If you have any further requests, please do not hesitate to contact us so that we can address your further concerns.

 

Reviewer #4: 

1. The language and the formatting of the article needs major revisions.

Response:

Thank you very much for your kind, thoughtful and valuable comments. We have reviewed the original text and made changes in language and format.

2. The author needs to briefly introduce a terminology when used for the first time in the article, for example - PM 2.5, moderate development, dot-like development etc.

Response:

Thank you very much for your kind, thoughtful and valuable comments. In the revised manuscript, every terminology is explained when they are mentioned for the first time, and the abbreviation for subsequent text is used.

3. In the introduction section, from the second paragraph onward coherence between sentences seem lacking. Even though each sentence adds some new information but the general flow of thought seems to be lacking in the introduction.

Response:

Thank you very much for your kind, thoughtful and valuable comments. Thanks for the referee’s suggestion. We have revised the introduction to make the sentences more fluent and the meaning expressed more complete and fluent. Please see pages 2-4 of the revised manuscript. 

…

Since 1972, Environmental problems have become global problems. As an important global Coordination Conference on environmental governance, United Nations Conference on environment and development (UNCED) is held to discuss global environmental issues and release corresponding policy documents. The countries all over the world have put forward the goal of sustainable development and participated in the process of environmental governance. The developed countries such as the United States have gone through the stage of industrialization and environmental problems have changed from pollution control to environmental behaviour governance. However, China is in the process of development, and its environmental pollution is becoming more and more serious. With the rapid growth of the economy and urbanization, air pollution has become a serious issue in China. Since 2013, China has continuously suffered from serious PM2.5 pollution with the average level of fine particulate matter (PM2.5) reaching 72μg/m3; 99.6% of the Chinese population lived in areas with PM2.5 exceeding the World Health Organization Air Quality Guideline of 10μg/m3. According to the Asian Development Bank Annual Report in 2012, less than 1% of China’s 500 largest cities had air quality up to the standards set by the World Health Organization; seven Chinese cities were listed among the ten most polluted cities in the world. Air pollution becomes not only a major long-term burden on the Chinese public but also a main obstacle to sustainable development.

The stress on the environment, society and resources are closely related to land use and economic activity. To solve the increasingly urgent problem of environmental protection, China's State Council issued the National Major Functional Zone Planning in 2010. Based on population distribution, land use, economic development and urbanization patterns, and the development potential and priorities in different regions, the country has been divided into four different functional zones, i.e. optimal development zones, key development zones, restricted development zones and non-development zones. The restricted development zones refer to the zones with weak carrying capacity of resources, poor conditions of large-scale agglomeration economy and population, and is related to the ecological security of the whole country or a large region; the prohibited development zones refer to all kinds of nature protection areas established according to law. Restricted development zones and the prohibited development zones mainly include natural forest protection zones, grassland degradation zones, natural disaster-prone zones, rocky desertification and desertification zones, and zones s with serious soil erosion. According to the requirements of National Major Functional Zone Planning, the four function zones should be all adhere to the priority of environmental protection and ecological restoration.

…

4. The key research questions that the article is trying to address should be mentioned explicitly at the end of introduction section.

Response:

Thank you very much for your kind, thoughtful and valuable comments. We have revised the introductory part and put forward the key research issues in the last paragraph of the introduction section.

5. Description of the content of the article need not be mentioned in the article.

Response:

Thank you very much for your kind, thoughtful and valuable comments. Thanks for the referee’s kind advice. We have deleted the description of the content as requiring.

6. In the 'Institutional Background' section, the author might want to give a detailed account of how are NKEFZs different from other regions as it will help the reader to understand the context.

Response:

Thank you very much for your kind, thoughtful and valuable comments. The manuscript provides related introductions to NKEFZs in the Introduction section and Institutional background section, but it is not detailed enough. Based on your suggestions, we re-describes NKEFZs, and introduce in detail the differences from other regions in terms of development intensity, industry guidance, ecological assessment, supervision, and compensation.

7. Since, the DID model detailed in the manuscript is essentially a regression model, the author should also talk about the assumptions of regression and how they were dealt with.

Response:

Thank you very much for your kind, thoughtful and valuable comments. We have added an explanation of the 5 main assumptions of the regression model in the Difference-in-differences approach section. Details as follows:

Assumption 1. The explanatory variable is a deterministic variable, not a random variable.

Assumption 2. The random error term has the property of zero mean.

Assumption 3. The random error term has the property of homoscedasticity.

Assumption 4. The random error term is not serially correlated.

Assumption 5. The random error term is not related to the explanatory variable.

For Assumption 1, the core variables of this article are derived from the government's plan for NKEFZ, and other control variables are obtained from the China Statistical Yearbook for Regional Economy or from the Chinese County Statistical Yearbook. They are all historical data that have already occurred, so the explanatory variables are all deterministic variables rather than random variables, so Assumption 1 holds. 

For Assumption 2, we first read a lot of previous related theoretical studies and set up the correct model so that Yi fluctuates up and down its expected value E(Yi), and the probability of occurrence of random disturbance items is the same, which can cancel each other. Therefore, Assumption 2 holds.

For Assumption 4, the model includes all the main variables related to the research topic of this article, so the interference factors are completely random, independent and uncorrelated, so Assumption 4 is established.

Furthermore, for Assumption 3, this paper takes the logarithm of all the control variables and the explained variables, so that the macroeconomic variables that originally have a right-skewed (or gradually increasing variance) nature become the same degree of dispersion relative to their respective mean. Therefore, Assumption 3 holds.

Finally, for Assumption 5, in the regression analysis, the explanatory variable X is a fixed value in repeated sampling and is a definite variable, while the random error term is random and belongs to a random variable, so the independent variable is not correlated with the random error term. Therefore, Assumption 5 holds.

8. The author should explain why the natural logarithm of PM 2.5 is taken as dependent variable and not the actual PM 2.5 itself.

Response:

Thank you very much for your kind, thoughtful and valuable comments. The reason for the logarithm of PM 2.5 is that the variation of the PM2.5 concentrations in each county is relatively large. The PM2.5 concentrations in the seriously polluted county reaches 70 ug/m3, while the low is only about 20. Such data differences can lead to heteroscedasticity problems. After taking logarithm, the heteroscedasticity problem can be avoided. The improvement percentage of PM2.5 (if the estimation coefficient is negative) can be obtained. In the research of economics, if the economic variables are absolute values, they are generally logarithmic, such as per capita GDP.

9. The parallel trend test appears to have been applied incorrectly. The correct method should be testing for difference in PM 2.5 between treatment and control for the years 2008, 2009, 2010, and 2011 individually. Ideally if the data conforms to the parallel trends assumption then the interaction term eco*year (eg. eco*2008) should not be statistically significant.

Response:

Thank you very much for your kind, thoughtful and valuable comments. The parallel trend test is basically satisfied, according to the conclusion of the time trend part. The article is equivalent to using the method of placebo test: assuming that the policy impact is in 2011, the treatment group is 1 and the others are 0 (eco*2011). The results show that the policy shocks are not significant before 2011, but are significantly negative only after 2012, indicating that policy shocks only have effect after 2012.

10. DID models can be easily interpreted using graphs/charts showing the deviance in the treatment line. The author might want to charts in addition to the tables.

Response:

Thank you very much for your kind, thoughtful and valuable comments. Indeed, parallel trend tests can be presented graphically. In this paper, regression table is used to present the results of time dynamic analysis (Table 6), which is equivalent to the parallel trend test.

11. The EKC hypothesis uses the economic growth as the independent variable and the environmental quality as the dependent variable. However, in the section 3.2.3 the author has concluded that establishment of NKEFZ (independent variable) has lead to improved economic development (dependent variable). Therefore, the conclusion doesn't exactly conform to the EKC hypothesis. Hence, the author should either avoid explaining the model from the EKC hypothesis lens or the author should provide valid arguments with regards to inverse relationship of the EKC curve.

Response:

Thank you very much for your kind, thoughtful and valuable comments. In our benchmark model, our explanatory variable is the degree of environmental pollution, and the level of economic development is one of the explanatory variables, so it meets the model setting of EKC. Only in the mechanism part, we take the level of economic development as the explanatory variable and the establishment of NKEFZ as the explanatory variable. This model does not test EKC. So our model is not contradictory. It is also possible that the conclusion of our study is inconsistent with EKC, since the county administrative region is taken as the research sample.

12. At more than one place in the results section, significance level is incorrectly mentioned as 'statistical level'.

Response:

Thank you very much for your kind, thoughtful and valuable comments. We have reviewed the original text and corrected this problem.

13. The author should interpret the results in the manuscript at a pre-defined significance level. At present, results have been interpreted at different significance level (1%, 5%, and 10%).

Response:

Thank you very much for your kind, thoughtful and valuable comments. The definition of significance level in the manuscript is a writing error. We have reviewed the manuscript and corrected this problem. We believe that it is acceptable to be significant at the 10% significance level, otherwise, the coefficient obtained is not significant. 

14. In table 4, which of the variables are dependent and independent are unclear.

Response:

Thank you very much for your kind, thoughtful and valuable comments. In Table 4, the explained variables are lnavgpm, lnmedpm and lnmaxpm. All the control variables are controlled in all matching methods.

15. The paragraph preceding table 5 needs significant revision.

Response:

Thank you very much for your kind, thoughtful and valuable comments. We have made a significant revision of the paragraph preceding table 5, and also made serious revisions to other parts of the full text. Please see page 19 of the revised manuscript. 

…

In order to compare the results with the benchmark regression, Table 5 presents the DID regression results after applying the matched samples. in Column (1) and (2), 14022 samples were matched, and the matching rate was 99%. in Column (3) -(6), The matched samples were relatively few, but the matching rate was above 98%. In the six models, the coefficients of eco*T were significantly negative, ranging from 0.1486 to 0.1700, which indicates that the establishment of NKEFZ may significantly reduce 14.86%-17% of the average concentration of PM2.5 pollution, the median value of PM2.5 pollution and the maximum concentration of PM2.5 pollution. The estimated results of PSM-DID model still do not support the Environmental Kuznets Curve hypothesis. The proportion of industrial industry and agricultural industry have significant positive and negative effects on PM2.5 pollution respectively.

…

16. In section 4.5, the in the first paragraph table 6 is mentioned instead of table 7.

Response:

Thank you very much for your kind, thoughtful and valuable comments. This problem is mainly caused by our writing errors. We have revised section 4.5 (Mechanism analysis).

17. In the conclusion section, the author should discuss about the implication and application of the research. How these results are useful for other geographies. The conclusion section should also talk about the limitations of the study and should provide recommendations to conduct similar studies in future.

Response:

Thank you very much for your kind, thoughtful and valuable comments. We have revised the conclusion section and added content such as the implication and application of the research, limitations and future studies in accordance with the reviewers’ suggestions. Please see pages 23-24 of the revised manuscript. 

…

The significance of this paper is that, on the one hand, it has confirmed that NKEFZ can indeed reduce air pollution, which provides a policy basis for the implementation of China's NKEFZ and other related policies. This paper also provides another governance model for China to further control air pollution and achieve high-quality economic growth. On the other hand, this paper not only provides experience and reference for other developing countries in the treatment of air pollution, but also provides a basis for policy comparison in developed countries such as the United States.

There are a few limitations to the study that are worth emphasizing. On the one hand, this paper only studies the impact of NKEFZ on PM2.5, and lacks research on changes in other air pollutants. On the other hand, this paper only studies the effect in the Chinese scenario, and lacks comparative studies with other developed and developing countries. The problem will leave this for future research.

…

Finally, thank you for your suggestions. All of your suggestions are very important. They are of important guiding significance for my thesis writing and scientific research. We have done our best to revise the manuscript, but if any additional revision is needed, we will certainly do so under your directions.

---

## [Decision Letter · Decision Letter 1]

6 Jan 2021

PONE-D-20-16975R1

Can the Establishment of National Key Ecological Functional Zones Improve Air Quality?

——An Empirical Study from China

PLOS ONE

Dear Dr. Li,

Thank you for submitting your manuscript to PLOS ONE. After careful consideration, we feel that it has merit but does not fully meet PLOS ONE’s publication criteria as it currently stands. Therefore, we invite you to submit a revised version of the manuscript that addresses the points raised during the review process.

We look forward to receiving your revised manuscript.

Kind regards,

Bing Xue, Ph.D.

Academic Editor

PLOS ONE

Reviewers' comments:

Reviewer's Responses to Questions

**Comments to the Author**

1. If the authors have adequately addressed your comments raised in a previous round of review and you feel that this manuscript is now acceptable for publication, you may indicate that here to bypass the “Comments to the Author” section, enter your conflict of interest statement in the “Confidential to Editor” section, and submit your "Accept" recommendation.

Reviewer #2: All comments have been addressed

2. Is the manuscript technically sound, and do the data support the conclusions?

Reviewer #2: Yes

3. Has the statistical analysis been performed appropriately and rigorously? 

Reviewer #2: Yes

4. Have the authors made all data underlying the findings in their manuscript fully available?

Reviewer #2: No

5. Is the manuscript presented in an intelligible fashion and written in standard English?

Reviewer #2: No

6. Review Comments to the Author

Reviewer #2: I read the paper. The paper has improved as compared to earlier version. I thank the authors for their effort. However, the paper can further be improved. The comments are listed below.

a) Although write-up has improved; the paper would benefit from one more detailed English edit for clarity and typos. For example; this sentence in line 96-97 can be improved “It is important to examine the effectiveness of the establishment of ecological functional zones in reducing air pollution because pollution has detrimental impacts on quality of life.” The last sentence reads “The problem will leave this for future research.”. I suggest a serious English edit.

b) The caption in figure 1 and Note needs revision. The author can follow the map caption in other papers.

c) In table-2: the author didn’t report the effect of the policy variable— ‘eco’ also. Is it intentionally removed? I suggest to include it.

d) The Authors has explained the reason for not presenting the DID estimates in the figure. Their argument is the estimates and parallel trend test provides the same information. However, the use of the figure will make the result visible and easily understandable.

e) The author argues the result in Table -5 is comparable to the Main result in table-2. When 99% sample is matched, it is unsurprising to see the same result. The question is ‘What does 99% sample matched imply?’ Are the treated and Not-treated counties the same? and if yes on what basis. The author has to mention the variable used to match the sample. Also, because the Table-4 result varies widely for a different method of matching and with the main result.

7. PLOS authors have the option to publish the peer review history of their article (what does this mean?). If published, this will include your full peer review and any attached files.

Reviewer #2: No

---

## [Author Response · Author response to Decision Letter 1]

10 Jan 2021

Response to Reviewers

First of all, we would like to thank reviewers for your insightful, constructive and helpful comments on our manuscript entitled “Can the Establishment of National Key Ecological Functional Zones Improve Air Quality? —— An Empirical Study from China”. We have carefully considered and addressed all the comments and made necessary revisions in the revised manuscript. We provide a point-by-point response to the reviewers’ comments below.

The points raised by the reviewers are written in bold font, whereas our responses are shown in normal font, and the quotation of the revised manuscript is shown in italic font. 

Reviewer #2:

I read the paper. The paper has improved as compared to earlier version. I thank the authors for their effort. However, the paper can further be improved. The comments are listed below.

Response:

Thank you again for your valuable suggestions and opinions on the improvement of the paper, which plays an important role in improving the quality of the paper. We will continue to revise the paper at your suggestion.

a) Although write-up has improved; the paper would benefit from one more detailed English edit for clarity and typos. For example; this sentence in line 96-97 can be improved “It is important to examine the effectiveness of the establishment of ecological functional zones in reducing air pollution because pollution has detrimental impacts on quality of life.” The last sentence reads “The problem will leave this for future research.”. I suggest a serious English edit.

Response:

Thank you for your suggestions on the English expressions in the manuscript. Indeed, there are many irregularities in our English expression. In the revised manuscript, we have asked English touch up company to make a comprehensive touch up of the voice, so as to meet the publishing requirements.

b) The caption in figure 1 and Note needs revision. The author can follow the map caption in other papers.

Response:

Thank you for pointing out that the title and annotation of Figure 1 are not standard. We modify the caption and Note of Figure 1. Please see Pages 4-5 of the revised manuscript:

…

The difference-in-differences estimator was the independent variable ( ). Since the effect of the eco-functional zone policy put into effect in June 2011and has a time lag, 2012 is chosen to reflect the impacts of the policy. The green areas in Figure 1 represent the NKEFZ, which are the treatment group in this paper; the red areas are the control group which represent the non-NKEFZ. These are 436 county-level administrative regions of the more than 2000 counties in China which belong to the NKEFZ. 

DID variables include the time dummy and the region dummy: the time dummy variables equal to 1 after the promulgation of the National Key Ecological Function Zones policy; the region dummy variables equal to 1 if the regions belong to the NKEFZ. Based on this, the interactive items between two dummy variables is DID variable. If the coefficient of the interactive items is negative and significant, it indicates that the air quality of the NKEFZ is improved.

Fig 1. The map of the NKEFZ and the non-NKEFZ in China.

Notes: The map is completed by ArcGIS software. The green areas are the treatment group which the counties belong to the NKEFZ, and the red areas is the control group which the counties belong to the non-NKEFZ. As some counties have no data, they are not included in the study sample, where these areas are shown in white on the map.

…

c) In table-2: the author didn’t report the effect of the policy variable— ‘eco’ also. Is it intentionally removed? I suggest to include it.

Response:

Thank you very much for questioning the problems in the regression results. In the regression, the coefficient of eco is not reported. The reason is that in the regression, the fixed effect of time and fixed effect of region are controlled, which leads to the multi-collinearity of eco and time effect. As a result, the coefficient cannot be estimated (the estimated coefficient is 0), so the coefficient is not reported in the regression. In the revised manuscript, we modified the notes for each table.

d) The Authors has explained the reason for not presenting the DID estimates in the figure. Their argument is the estimates and parallel trend test provides the same information. However, the use of the figure will make the result visible and easily understandable.

Response:

Thank you for your advice. In the revised manuscript, we made a parallel trend test figure based on the results of table 6, and revised the corresponding content of the paper. Please see Pages 4-5 of the revised manuscript:

…

The estimated results in columns (2), (4) and (6) of table 6 are made into Figure 2. It can be seen that after controlling for other factors, the three explained variables are significantly negative at the level of 5% after 2012, but not significant before 2011, which indicates that the three estimates meet the parallel trend test.

Fig. 2 Parallel trend tests. Those figures are based on the estimated results in columns (2), (4) and (6) of table 6, with a confidence level of 5%.

…

e) The author argues the result in Table -5 is comparable to the Main result in table-2. When 99% sample is matched, it is unsurprising to see the same result. The question is ‘What does 99% sample matched imply?’ Are the treated and Not-treated counties the same? and if yes on what basis. The author has to mention the variable used to match the sample. Also, because the Table-4 result varies widely for a different method of matching and with the main result.

Response:

Thank you for your question. Your question is very good. This is also the confusion we face in the process of writing. First of all, the sample loss after matching is less, only 1-2%, indicating that some samples do not match the treatment group, and these samples may be the year area with more serious pollution. Secondly, the estimated results of PSM-DID are much different from those of the main model. We think that it is because there are no matched samples that the pollution degree of these areas (counties) is more serious. Generally speaking, the counties that can become the NKEFZ are not the most serious in pollution. Therefore, those with better matching are not the ones with the most serious pollution. After removing the most polluted samples (the estimated coefficient becomes smaller), the difference between the treatment group and the control group is not so big, but it is still significantly negative, indicating that the effect of the policy still exists.

Finally, thank you for your suggestions. All of your suggestions are very important. They are of important guiding significance for my thesis writing and scientific research. We have done our best to revise the manuscript, but if any additional revision is needed, we will certainly do so under your directions.

---

## [Editor Report · Decision Letter 2]

19 Jan 2021

Can the Establishment of National Key Ecological Functional Zones Improve Air Quality?

——An Empirical Study from China

PONE-D-20-16975R2

Dear Dr. Li,

We’re pleased to inform you that your manuscript has been judged scientifically suitable for publication and will be formally accepted for publication once it meets all outstanding technical requirements.

Kind regards,

Bing Xue, Ph.D.

Academic Editor

PLOS ONE
---

## [Editor Report · Acceptance letter]

3 Feb 2021

PONE-D-20-16975R2 

Can the Establishment of National Key Ecological Functional Zones Improve Air Quality? —— An Empirical Study from China 

Dear Dr. Li:

I'm pleased to inform you that your manuscript has been deemed suitable for publication in PLOS ONE. Congratulations! Your manuscript is now with our production department. 

Kind regards, 

on behalf of

Professor Bing Xue 

Academic Editor

PLOS ONE